# Evaporation of sulphate aerosols at low relative humidity

Georgios Tsagkogeorgas[1], Pontus Roldin[2,3], Jonathan Duplissy[2,4], Linda Rondo[5], Jasmin Tröstl[6], Jay G. Slowik[6], Sebastian Ehrhart[5a], Alessandro Franchin[2], Andreas Kürten[5], Antonio Amorim[7], Federico Bianchi[2], Jasper Kirkby[5,8], Tuukka Petäjä[2], Urs Baltensperger[6], Michael Boy[2], Joachim Curtius[5], Richard C. Flagan[9], Markku Kulmala[2,4], Neil M. Donahue[10], Frank Stratmann[1]

[1]Leibniz Institute for Tropospheric Research, 04318, Leipzig, Germany
[2]Department of Physics, University of Helsinki, P.O. Box 64, 00014, Helsinki, Finland
[3]Division of Nuclear Physics, Lund University, P.O. Box 118, 221 00, Lund, Sweden
[4]Helsinki Institute of Physics, University of Helsinki, P.O. Box 64, 00014 Helsinki, Finland
[5]Institute for Atmospheric and Environmental Sciences, Goethe University Frankfurt, 60438, Frankfurt am Main, Germany
[6]Paul Scherrer Institute, CH–5232, Villigen, Switzerland
[7]Fac. Ciencias & CENTRA, Universidade de Lisboa, Campo Grande, 1749–016, Lisboa, Portugal
[8]CERN, CH–1211, Geneva, Switzerland
[9]California Institute of Technology, Pasadena, CA 91125, USA
[10]Center for Atmospheric Particle Studies, Carnegie Mellon University, Pittsburgh, PA 15213, USA
[a]now at: Atmospheric Chemistry Department, Max Planck Institute for Chemistry, 55128, Mainz, Germany

*Correspondence to*: Georgios Tsagkogeorgas (george.tsagkogeorgas@tropos.de)

**Abstract.** Evaporation of sulphuric acid from particles can be important in the atmospheres of Earth and Venus. However, the equilibrium constant for the dissociation of $H_2SO_4$ to bisulphate ions, which is the one of the fundamental parameters controlling the evaporation of sulphur particles, is not well constrained. In this study we explore the volatility of sulphate particles at very low relative humidity. We measured the evaporation of sulphur particles versus temperature and relative humidity in the CLOUD chamber at CERN. We modelled the observed sulphur particle shrinkage with the ADCHAM model. Based on our model results, we conclude that the sulphur particle shrinkage is mainly governed by $H_2SO_4$ and potentially to some extent by $SO_3$ evaporation. We found that the equilibrium constants for the dissociation of $H_2SO_4$ to $HSO_4^-$ ($K_{H_2SO_4}$) and the dehydration of $H_2SO_4$ to $SO_3$ ($^xK_{SO_3}$) are $K_{H_2SO_4}=2–4\cdot10^9$ mol·kg$^{-1}$ and $^xK_{SO_3}\geq1.4\cdot10^{10}$ at 288.8±5 K.

Key words: sulphate aerosol evaporation, sulphuric acid dissociation, sulphuric acid equilibrium constants, sulphuric acid vapour pressure, water activity, activity coefficients, Earth's and Venus' stratospheres, ADCHAM, CLOUD experiment

## 1 Introduction

Suspended particulate matter in the atmosphere plays a key role in Earth's climate. Atmospheric aerosol particles affect the amount of solar radiation absorbed by the Earth system. This is accomplished either when atmospheric aerosol particles directly absorb or scatter incoming solar energy (causing warming or cooling), or when particles act as cloud condensation or ice nuclei

(leading to an increase in cloud albedo, which causes cooling). A substantial fraction of particle number and mass across a wide range of environmental conditions arises from sulphur emissions (Clarke et al., 1998a; Turco et al., 1982).

Sulphur in Earth's atmosphere in turn originates from natural phenomena like volcanic eruptions and biota decomposition. Violent volcanic eruptions can loft sulphur dioxide ($SO_2$) to the stratosphere, which can then form sulphur aerosol particles. Those sulphur aerosols can remain suspended in the stratosphere for ~1–2y before falling into the troposphere (Wilson et al., 1993; Deshler, 2008). The three main natural agents for sulphate aerosol formation in troposphere are dimethyl sulphide (DMS), which arises from marine phytoplankton decomposition (Charlson et al., 1987; Kiene, 1999; Simó and Pedrós–Alió, 1999), $SO_2$, which occurs naturally as a decay product of plant and animal matter (Grädel and Crutzen, 1994; Hübert, 1999; Capaldo et al., 1999), and carbonyl sulphide (OCS), which is emitted from anaerobic biological activity and provides the main non–volcanic flux of sulphur into the stratosphere (Galloway and Rodhe, 1991; Rhode, 1999).

The atmospheric sulphate burden is substantially perturbed by sulphur emissions associated with anthropogenic activities. The largest anthropogenic source of sulphur is fossil–fuel combustion; coal is the predominant source, but also heavy fuel oil is important (Öm et al., 1996; Smith et al., 2001). Fossil–fuel combustion constitutes ~⅔ of the total global sulphur flux to the atmosphere (Rhode, 1999; Wen and Carignan, 2007), and dominates emissions in most populated regions. Other anthropogenic factors also affect the sulphuric acid ($H_2SO_4$) budget, notably sulphur aerosol formation in aircraft plumes (Fahey et al., 1995; Curtius et al., 1998), and extensive sulphur use in industry with a direct environmental impact on local scale. However, on a regional to global scale the acidification of fresh water and forest ecosystems is mainly caused by wet and dry deposition of $SO_2$ and sulphate particles (Simpson et al., 2006).

Sulphur is also a crucial constituent in Venus' atmosphere, an environment with very low relative humidity (RH) (Moroz et al., 1979; Hoffman et al., 1980a), forming the main cloud layer in the form of sulphuric acid droplets (Donahue et al., 1982), which are maintained in an intricate photochemical cycle (photooxidation of OCS, Prinn 1973). Sulphuric acid's reaction paths remain a subject of investigation (Zhang et al., 2010), which makes the study of the sulphur cycle (including the sulphur species SO, $SO_2$, $SO_3$, $H_2SO_4$) an important endeavour for understanding both the chemistry and climate of Venus (Mills et al., 2007; Hashimoto and Abe, 2000).

$H_2SO_4$ serves as an effective nucleating species and, thus, strongly influences atmospheric new–particle formation (Laaksonen and Kulmala, 1991; Weber et al., 1999; Kulmala et al., 2000; Yu and Turco, 2001; Fiedler et al., 2005; Kuang et al., 2008). The nucleation rate, which is the formation rate ($cm^{-3} \cdot s^{-1}$) of new particles at the critical size, strongly depends upon the saturation ratio of $H_2SO_4$. Uncertainty in this ratio results in an uncertainty of several orders of magnitude in the calculated nucleation rate (Roedel, 1979). To model the excess $H_2SO_4$ responsible for the gas–to–particle conversion it is necessary to know the vapour pressure of $H_2SO_4$ over sulphuric acid and/or neutralized solutions.

The sulphuric acid vapour pressure appears through the free–energy term in the exponent of the new–particle formation rate (Volmer and Weber, 1926; Stauffer, 1976). Quantitative theoretical predictions of nucleation rates are highly uncertain because the pure $H_2SO_4$ equilibrium vapour pressure is not well known (Gmitro and Vermeulen, 1964; Doyle, 1961;

Kiang and Stauffer, 1973). However, accurate calculations of the $H_2SO_4$ vapour pressure require accurate equilibrium rate constant values to constrain the reactions of formation and dissociation of $H_2SO_4$ in aqueous solutions.

While $H_2SO_4$ is often presumed to be practically non–volatile, this is not always the case. There are several circumstances on Earth and Venus where the vapour pressure of $H_2SO_4$ matters; specifically, at very low RH, high temperature (T), when there is a deficit of stabilizing bases, and when particles are very small. A very important region of Earth's environment is the upper stratosphere where these conditions prevail (Vaida et al. 2003). Under these conditions $H_2SO_4$ can evaporate from particles. This can either inhibit growth of nanoparticles or lead them to shrink.

Furthermore, molecular $H_2SO_4$ is never the dominant constituent in sulphuric acid solutions. It will completely dehydrate to sulphur trioxide ($SO_3$, which is extremely volatile) in a truly dry system and yet almost entirely dissociate into bisulphate ion ($HSO_4^-$) and hydronium cation ($H_3O^+$) in the presence of even trace water ($H_2O$) (Clegg and Brimblecombe, 1995). This is why $H_2SO_4$ is such a powerful desiccant. Also, bases such as ammonia ($NH_3$) will enhance chemical stabilization and form sulphate salts. The thermodynamics of the $H_2SO_4$–$H_2O$ system at low RH are uncertain, so we seek to improve our understanding of this part of the phase diagram. To accomplish this, we measured the shrinkage of nearly pure $H_2SO_4$ particles in the CLOUD chamber at CERN at very low RH and then simulated these experiments with an aerosol dynamics model coupled with a thermodynamics model to constrain the equilibrium constants, for the dissociation $K_{H_2SO_4}$ and the dehydration $^xK_{SO_3}$, of $H_2SO_4$ coupling $HSO_4^-$, $H_2SO_4$, and $SO_3$. These new values can be used in models that simulate the evolution of sulphate aerosol particles in the atmospheres of Venus and Earth.

## 2 Aqueous phase sulphuric acid reactions

$H_2SO_4$ dissociation and potential dehydration to $SO_3$ are the principal subjects of this study. In aqueous solutions $H_2SO_4$ can dissociate in two steps.

$$H_2SO_{4(aq)} \xleftrightarrow{K_{H_2SO_4}} HSO_{4(aq)}^- + H^+ \tag{R1}$$

$$HSO_{4(aq)}^- \xleftrightarrow{K_{HSO_4^-}} SO_{4(aq)}^{2-} + H^+ \tag{R2}$$

$H_2SO_4$ partially dissociates to form $HSO_4^-$ via reaction 1 (R1). $K_{H_2SO_4}$ represents the equilibrium constant for R1. $HSO_4^-$ can then undergo a second dissociation reaction (R2) to form a sulphate ion ($SO_4^{2-}$). In above reactions, sulphur's oxidation number is 6 (S(VI)).

For dilute aqueous solutions, R1 is considered to be complete. However, when the mole fraction of S(VI) exceeds ~0.5, $H_2SO_4$ can be detected in the solution (Walrafen et al., 2000; Margarella et al., 2013). When $H_2SO_4$ is present in the solution, dehydration of $H_2SO_4$ to form $SO_3$ (R3) can also be important (Wang et al., 2006; Que et al., 2011). $^xK_{SO_3}$ represents the equilibrium constant for R3 on a mole fraction basis.

$$SO_{3(aq)} + H_2O \xleftrightarrow{^xK_{SO_3}} H_2SO_{4(aq)} \tag{R3}$$

NH$_3$, which mainly originates from anthropogenic agriculture emissions, is the most abundant base in atmospheric secondary aerosol particles. NH$_3$ neutralises sulphuric acid particles by reacting with H$^+$ and forming an ammonium ion (NH$_4^+$) (R4).

$$\text{NH}_{3(aq)} + \text{H}^+ \xleftarrow{\quad K_{NH_3} \quad} \text{NH}_4^+ \tag{R4}$$

Even in the cleanest environments, such as the stratosphere, NH$_3$ is present at low concentrations and NH$_{3(g)}$ will be dissolved in the acidic sulphate particles.

## 3 Methods

In the CLOUD (Cosmics Leaving OUtdoor Droplets, Kirkby et al. (2011)) chamber at CERN, we measured the H$_2$SO$_4$ aerosol evaporation process under precisely controlled temperature and relative humidity. We designed experiments to accomplish a

gradual decrease of RH (from 11.0 to 0.3 %) under atmospherically relevant conditions. To understand the processes governing the measured particle evaporation, we modelled the experiments with the Aerosol Dynamics, gas– and particle–phase chemistry model for laboratory CHAMber studies (ADCHAM, Roldin et al., 2014).

### 3.1 Experimental set up

Details of the CLOUD chamber, the main element of the experimental set up can be found in Kirkby et al. (2011) and Duplissy

et al. (2016). For the experiments described here, we formed and grew sulphuric acid particles in the chamber by oxidising SO$_2$ with OH radicals that were generated by photolysing O$_3$ and allowing the resulting O($^1$D) to react with water vapour. During these experiments we fed the aerosol population to an array of instruments for characterisation of both physical and chemical properties.

We utilized the following instruments to measure gas-phase concentrations: a SO$_2$ monitor (Enhanced Trace Level

SO$_2$ 15 Analyser, Model 43i–TLE, Thermo Scientific, USA), an O$_3$ monitor (TEI 49C, Thermo Environmental Instruments, USA) and a Chemical Ionisation Mass Spectrometer (CIMS) measured the gas–phase H$_2$SO$_4$ concentration ([H$_2$SO$_{4(g)}$] between ~$5 \cdot 10^5$ and ~$3 \cdot 10^9$ cm$^{-3}$, Kürten et al., 2011; Kürten et al., 2012). The CIMS data provided the total gaseous sulphuric acid concentration, [H$_2$SO$_{4(g)}$] without constraining the hydration state of the evaporating molecules (e.g. H$_2$SO$_4$ associated with one, two, or three H$_2$O molecules).

We measured the evolution of the aerosol number size distribution with a Scanning Mobility Particle Sizer (SMPS, Wang and Flagan, 1990), which recorded the dry particle mobility diameter in the size range from about 10 to 220 nm. We operated the SMPS system with a recirculating dried sheath flow (RH<14 % controlled by a silicon dryer) with a sheath to aerosol sample flow ratio of 3:0.3 L. We maintained the Differential Mobility Analyser (DMA) and recirculating system at 278–288 K by means of a temperature control rack, while we operated the Condensation Particle Counter (CPC) at room

temperature. We corrected the SMPS measurements for charging probability, including the possibility of multiple charges, diffusion losses, and CPC detection efficiency.

We measured aerosol particle chemical composition with an Aerodyne Aerosol Mass Spectrometer (AMS) quantifying sulphate, nitrate, ammonium and organics for particles between 50 and 1000 nm aerodynamic diameter (Jimenez et al., 2003a; Drewnick et al., 2006; Canagaratna et al., 2007). The AMS provided the mass concentration measurements ($\mu g \cdot m^{-3}$) calculated from the ion signals by using measured air sample flow rate, nitrate ionization efficiency (IE) and relative IE of the other species.

## 3.2 The experimental procedure

To study aerosol particle evaporation, the formation of sulphuric acid particles preceded. At the lowest $H_2O$ levels (*RH<11 %*) and in the presence of $O_3$, controlled UV photo–excitation reactions initiated the oxidation of $SO_2$ to $H_2SO_4$. Sulphuric acid particles nucleated and grew to a size of ~220 nm by condensation of $H_2SO_{4(g)}$ at a quasi–constant gas phase concentration (~$1 \cdot 10^9$ $cm^{-3}$ with an uncertainty of >20 %). The $H_2SO_4$ formation and particle growth ended when we closed the shutters in the front of the UV light source. Afterwards, we induced particle shrinkage by decreasing the RH. We decreased the RH in two separate ways; either by minimizing the influx of water vapour to the chamber, or by increasing the temperature. This separation in experimental procedures gave the ability to achieve and control extremely low RH values (Table 1).

After the end of the particle formation period and during the initial steps of evaporation, before the RH started to decrease, the aerosol size distribution remained nearly constant. Subsequently, the RH decreased gradually initiating the particle evaporation. When the RH reached a certain low value (*RH≤1.5 %* for *T=288.8* K) the particles shrank rapidly, as revealed by the SMPS measurements, and the [$H_2SO_{4(g)}$] increased until it reached a peak value (see Supplement, Fig. S1). The [$H_2SO_4$]$_{peak}$ was significantly higher than the background concentration before the onset of evaporation (Table 1). After reaching a maximum in gas–phase concentration, the sulphuric acid decreased again, though the size distribution remained stable (e.g., ~50 (±10) nm for experiments 1 and 2, see Sect. 4.3) depending on the RH and T conditions. This behaviour revealed that the remaining aerosol could not be pure sulphuric acid, but rather consisted of a more stable chemical mixture that inhibited further evaporation.

Similarly, the AMS recorded the evaporation of particles (see Supplement, Fig. S1). The AMS measurements showed that the particles were composed almost exclusively of sulphuric acid (but not pure $H_2SO_4$). Based on AMS data, calculations of the kappa value ($\kappa$), which is defined as a parameter that describes the aerosols water uptake and cloud condensation nucleus activity (CCN activity), (Petters and Kreidenweis, 2007) of the mixed particles as a function of time during particle evaporation (see Supplement, Fig. S2) yield a value close to the $\kappa$ for pure sulphuric acid particles (Sullivan et al., 2010). A $\kappa$ value is indicative of the solubility of aerosol particles, with *κ=0* referring to an insoluble particle and *κ=0.7* to pure sulphuric acid particles. $\kappa$ is computed by the approximate equation, Eq. (1)

$$\kappa = \frac{4 \cdot A^3}{27 \cdot D_d^3 \cdot ln^2 S_c} \tag{1}$$

when the critical diameter $D_d$ and critical saturation $S_c$ (or supersaturation, $s_c$, when referring to CCN activity) are known. The term A can be calculated from the water properties.

## 3.3 The model framework

In the present work we use ADCHAM (Roldin et al., 2014, 2015) to study the evolution of the particle number size distribution and particle chemical composition. Instead of simulating the new–particle formation in the CLOUD chamber, we use the measured particle number size distribution before the UV–lights are turned off as well as time sequences of RH, T and $[H_2SO_{4(g)}]$ as inputs to the model (Fig. 1). In order to capture the evolution of the particle number size distribution we consider Brownian coagulation, particle wall deposition, condensation and evaporation of $H_2SO_4$, $SO_3$ and $H_2O$ from the particles.

### 3.3.1 The activity coefficients

Within an aqueous electrolyte solution, such as the $H_2SO_4$–$SO_3$–$H_2O$ system, cations, anions and molecular species all disrupt ideality. Here, we consider interactions between ions ($HSO_4^-$, $SO_4^{2-}$, $NH_4^+$, $H^+$) and molecules ($H_2SO_4$, $SO_3$, $H_2O$) in the particle–phase chemistry model. To calculate the molality based activity coefficients for the inorganic ions ($\gamma_i$) and the mole fraction based activity coefficient for water ($f_{H_2O}$) we apply the Aerosol Inorganic Organic Mixtures Functional groups Activity Coefficients (AIOMFAC) model (validated at room temperatures, Zuend et al., 2008 and 2011). The reference state for ions and water in the model is an infinitely dilute aqueous solution ($\gamma_i(\chi_{H_2O} \rightarrow 1)=1$ and $f_{H_2O}(\chi_{H_2O} \rightarrow 1)=1$.

For relatively dilute $H_2SO_{4(aq)}$ solutions (low solute concentration), typical for most atmospheric conditions, it is reasonable to assume that the dissociation of $H_2SO_4$ to $HSO_4^-$ (R1) is complete (Clegg et al., 1998, Zuend et al., 2008). However, in this work we demonstrate that this assumption fails at low RH and also for small particles with a large Kelvin term. Furthermore, at a very low water activity ($a_w$) (less than ~0.01) a non–negligible fraction of the $H_2SO_4$ could potentially decompose to $SO_3$ (R3); if this is the case, the thermodynamic model need to consider not only R1 but R3 as well (Fig. 1).

Since AIOMFAC does not consider inorganic non–electrolyte compounds like $H_2SO_4$ and $SO_3$ we implement additionally to this the symmetric electrolyte–NonRandom Two–Liquid (eNRTL) activity coefficient model (Bollas et al., 2008, Song and Chen, 2009) which is optimized for the $H_2SO_4$–$H_2O$–$SO_3$ systems by Que et al., (2011). In this work we use the regressed eNRTL binary interaction parameters from Que et al., 2011. Following the convention of the eNRTL model (Chen et al., 1982), we set the unknown binary parameters for $NH_4^+$–molecule, molecule–$NH_4^+$ and $NH_4^+$–ions to – 4, 8 and 0, respectively.

The reference state of the molecular species in eNRTL is defined as the pure liquid. eNRTL provides mole fraction based activity coefficients for $H_2SO_4$ and $SO_3$, $f_{H_2SO_4}$ and $f_{SO_3}$, respectively. ADCHAM calculates $f_{H_2SO_4}$ and $f_{SO_3}$ as a function of $a_w$ and $N{:}S$, $\chi_{N(-III)}{:}\chi_{S(VI)}$ (Fig. S3). The modelled $f_{H_2SO_4}$ and $f_{SO_3}$ approach unity not only at the standard state of the pure liquids ($f_{H_2SO_4}(\chi_{H_2SO_4} \rightarrow 1)=1$ and $f_{SO_3}(\chi_{SO_3} \rightarrow 1)=1$), but also for the infinitely dilute aqueous solution ($f_{H_2SO_4}(\chi_{H_2O} \rightarrow 1)=1$ and $f_{SO_3}(\chi_{H_2O} \rightarrow 1)=1$). This is because the eNRTL binary $H_2O$–$H_2SO_4$ and $H_2O$–$SO_3$ interaction parameters are zero

in the model. For all conditions between these limiting states, the short–range ion ($HSO_4^-$, $SO_4^{2-}$, $NH_4^+$, $H^+$) –molecule ($H_2SO_4$, $SO_3$) interactions, and Pitzer–Debye–Hückel long–range ion–molecule interactions influence the modelled $f_{H_2SO_4}$ and $f_{SO_3}$. At $T=288.8$ K, $f_{H_2SO_4}$ reaches the highest values ($\sim2.29$) when $a_w\approx0.25$ and $f_{SO_3}$ reaches the highest values ($\sim1.95$) when $a_w\approx0.35$ (Fig. S3). We also assume that the activity coefficient of $NH_3$ is unity for the model simulations. However, sensitivity tests performed for $\gamma_{NH_3}=0.1$ and $\gamma_{NH_3}=10$ reveal that, for the acidic particles ($N:S<1$), our model results are completely insensitive of the absolute value of $\gamma_{NH_3}$.

### 3.3.2 The particle phase composition

If ammonium cation ($NH_4^+$) is present in the sulphuric acid particles, then solid ammonium bisulfate ($NH_4HSO_4(s)$) may form when the S(VI) and $H_2O$ start to evaporate from the particles. However, the particles may also stay as highly supersaturated droplets with respect to the crystalline phase (Zuend et al., 2011). The particle number size distribution measurements in our experiments did not indicate a sudden drop in particle size during evaporation, which would be expected if the particles crystalized and all particle water was suddenly removed. Thus, in the present work we do not consider formation of any solid salts. We further neglect the influence of any mass–transfer limitations in the particle phase, and assume that the particle ion–molecule equilibrium composition (R1–R3) and water content can be modelled as equilibrium processes (because they are established rapidly compared to the composition change induced by the evaporation of $H_2SO_4$ and $SO_3$). We use the thermodynamic model to update the particle equilibrium water content, mole fractions and activity coefficients of all species. Then the model considers the gas–particle partitioning of $H_2SO_4$ and $SO_3$ with a condensation algorithm in the aerosol dynamics model (Sect. 3.3.5). The time step set in the model is 1s.

The thermodynamic model uses an iterative approach to calculate the particle equilibrium mole fractions of $H_2O$, $H_2SO_4$, $SO_3$, $HSO_4^-$, $SO_4^{2-}$, $NH_3$, $NH_4^+$ and $H^+$, based on the current time step, known RH, and absolute number of moles of S(VI) and N(–III) for each particle size bin. The modelled particle–phase mole fraction of N(–III) during the evaporation experiments is always substantially lower than that of S(VI) ($N:S<0.7$). For these particles the saturation vapour pressure of $NH_3$ is always less than $10^{-10}$ Pa, within the experimental water activity range 0–0.11 and $\gamma_{NH_3}\geq0.1$. Thus, it is reasonable to assume that during the experiments $NH_3$ does not evaporate from the particles.

Based on the particle diameters from the previous time step (which depend on the particle water content), the thermodynamic model starts by calculating $a_w$ for each particle size, considering the Kelvin effect. Given $a_w$, the model estimates the particle water mole fraction. Then the model calculates the $H^+$ molality in the aqueous phase via a 4th order polynomial, derived from the ion balance equation, Eq. (2) in combination with the thermodynamic equilibrium constant equations, Eq. (3–6), and the S(VI) and N(–III) mole balance equations, Eq. (7–8). The maximum positive real root of this polynomial gives the $H^+$ concentration, $[H^+]$.

$$\left[H^+\right]+\left[NH_4^+\right]=\left[HSO_4^-\right]+2\left[SO_4^{2-}\right] \tag{2}$$

$$K_{H_2SO_4} = \frac{[HSO_4^-] \cdot \gamma_{HSO_4^-} \cdot [H^+] \cdot \gamma_{H^+}}{[H_2SO_4] \cdot \gamma_{H_2SO_4}} \tag{3}$$

$$K_{HSO_4^-} = \frac{[SO_4^{2-}] \cdot \gamma_{SO_4^{2-}} \cdot [H^+] \cdot \gamma_{H^+}}{[HSO_4^-] \cdot \gamma_{HSO_4^-}} \tag{4}$$

$${}^x K_{SO_3} = \frac{\chi_{H_2SO_4} \cdot f_{H_2SO_4}}{\chi_{SO_3} \cdot f_{SO_3} \cdot \chi_{H_2O} \cdot f_{H_2O}} \tag{5}$$

$$K_{NH_3} = \frac{[NH_3] \cdot \gamma_{NH_3} \cdot [H^+] \cdot \gamma_{H^+}}{[NH_4^+] \cdot \gamma_{NH_4^+}} \tag{6}$$

$$n_{S(VI)} = n_{H_2SO_4} + n_{HSO_4^-} + n_{SO_4^{2-}} + n_{SO_3} \tag{7}$$

$$n_{N(-III)} = n_{NH_4^+} + n_{NH_3} \tag{8}$$

The thermodynamic equilibrium coefficients for $H_2SO_4$ and $HSO_4^-$ dissociations and $NH_3$ protonation (Eq. 3, 4 and 6) are given in a molality based form while the equilibrium coefficient in Eq. (5), which involves the equilibration between the different solvents ($H_2O$, $SO_3$ and $H_2SO_4$), is given in a mole fraction based form. The Eq. (5) is given in a mole fraction based

form for the following reasons: a) the eNRTL provides mole fraction based activity coefficients, and b) if Eq. (5) would be applied for $a_w$ that are even lower than considered in this work, the assumption of using molalities, i.e. where water is considered to be the only solvent, will not be acceptable. The model calculates $K_{HSO_4^-}$ and $K_{NH_3}$ (mol·kg$^{-1}$) with Eq. (9) and Eq. (10) (Jacobson, 2005a). We treat $K_{H_2SO_4}$ and ${}^x K_{SO_3}$ as unknown model fitting parameters.

$$K_{HSO_4^-} = 1.015 \cdot 10^{-2} \cdot e^{\left( 8.85 \cdot \left( \frac{298}{T} - 1 \right) + 25.14 \cdot \left( 1 + \ln\left( \frac{298}{T} \right) - \frac{298}{T} \right) \right)} \tag{9}$$

$$K_{NH_3} = 1.7882 \cdot 10^9 \cdot e^{21.02 \cdot \left( \frac{298}{T} - 1 \right)} \tag{10}$$

Once [H$^+$] is determined, all other ion and molecule concentrations can be derived from Eq. (2–8). Based on the new estimated particle–phase ion and molecule mole fractions, the thermodynamic model uses AIOMFAC and eNRTL to update the ion and molecule activity coefficients. The model then repeats the whole procedure iteratively until the relative change in the concentration and activity coefficients for each compound is less than 10$^{-9}$ between successive iteration steps. To stabilize

convergence, the model estimates activity coefficients used in the proceeding iteration as a weighted average of the values from the previous and present iteration time steps.

### 3.3.3 $H_2SO_4$ and $SO_3$ in the gas–phase

In the gas phase only a fraction of $H_2SO_4$ is in the form of pure sulphuric acid molecules while the rest of the $H_2SO_4$ is in a hydrated form. In this work we use the parameterization from Hanson and Eisele (2000), who measured the diffusion loss rate of $H_2SO_4$ to flow–tube walls at different RH, to estimate the RH–dependent effective diffusion coefficient of $H_2SO_{4(g)}$.

In the gas phase, $SO_3$ reacts rapidly with $H_2O$ to form $H_2SO_4$. Based on the measured loss rate of $SO_3$, which shows a second–order dependence on the water vapour concentration (Jayne et al., 1997), we estimate that $SO_3(g)$ is converted to $H_2SO_{4(g)}$ in less than 1s during the CLOUD chamber experiments, even at the lowest RH. Because of this rapid conversion to $H_2SO_4$ and the high vapour pressure of $SO_3$ (Eq. 12), it is reasonable to assume that the gas–phase concentration of $SO_3$ (vapour pressure, $p_{\infty,SO_3(g)}$) is negligibly low.

### 3.3.4 Saturation vapour pressures, surface tension and particle density

We use Eq. (11) and (12) to calculate the temperature dependent sub–cooled pure–liquid saturation vapour pressures for $H_2SO_4$ and $SO_3$ ($p_{0,i}$, where $i$ refers to $H_2SO_4$ or $SO_3$ in Pa). Equation (11) is based on the work of Ayers et al. (1980), with corrections for lower temperatures by Kulmala and Laaksonen (1990). We use the (best fit) $L$ parameter value of $-11.695$ (Noppel et al., 2002, Noppel–Kulmala–Laaksonen, N–K–L parameterisation, see Supplement Fig. S5 (a)). Equation (12) is based on the work of Nickless (1968) (see Supplement Fig. S5 (b)).

$$p_{0,H_2SO_4} = 101325 \cdot e^{\left( L+10156 \cdot \left[ \frac{1}{360.15} - \frac{1}{T} + \frac{0.38}{545} \cdot \left( 1+\ln\left( \frac{360.15}{T} \right) - \frac{360.15}{T} \right) \right] \right)} \tag{11}$$

$$p_{0,SO_3} = e^{\left( 28.9239 - \frac{7000}{T} \right)} \cdot 133.3224 \tag{12}$$

As an alternative to Eq. (11) and (12) we also use the $H_2SO_4$ and $SO_3$ pure–liquid saturation vapour pressure parameterisations from Que et al., 2011 (originally from the Aspen Plus Databank, Fig. S5).

We calculate the saturation vapour pressures of $H_2SO_4$ and $SO_3$ for each particle size with Eq. (13), using the mole fractions ($\chi_{i,j}$) and mole fraction based activity coefficients ($f_{i,j}$) of $H_2SO_4$ and $SO_3$ (from the thermodynamic model) and the Kelvin term, $C_{k,i,j}$ Eq. (14) for compound $i$ in particle size bin $j$.

$$p_{s,i,j} = p_{0,i} \cdot a_{i,j} \cdot C_{k,i,j} \tag{13}$$

where $a_{i,j} = \chi_{i,j} \cdot f_{i,j}$

$$C_{k,i,j} = e^{\left( \frac{4 \cdot M_i \cdot \sigma_j}{R \cdot T \cdot \rho_{p,j} \cdot D_{p,j}} \right)} \tag{14}$$

$a_{i,j}$ is the activity of compound $i$ in size bin $j$, $T$ is the temperature in Kelvin, $R$ is the universal gas constant (J·mol$^{-1}$·K$^{-1}$), $M_i$ is the molar mass (kg·mol$^{-1}$) of compound $i$, $\rho_{p,j}$ is the density (kg·m$^{-3}$) of the liquid particles, $\sigma_j$ is the surface tension (N·m$^{-1}$) and $D_{p,j}$ is the particle diameter (m) of the particles in size bin $j$.

As an alternative approach we also model the evaporation of $H_2SO_4$ using composition dependent $H_2SO_4$ activities $(\alpha_{H_2SO_4,j})$ derived directly from the tabulated values of the difference in chemical potentials between the sulphuric acid in aqueous solution and that of the pure acid $(\mu_{H_2SO_4,j} - \mu^0_{H_2SO_4})$. The tabulated values that are valid at 298.15 K are taken from Giauque et al. (1960). The relationship between $\mu_{H_2SO_4,j} - \mu^0_{H_2SO_4}$ and $\alpha_{H_2SO_4,j}$ is given by Eq. (15).

$$\ln(a_{H_2SO_4,j}) = (\mu_{H_2SO_4,j} - \mu^0_{H_2SO_4})/(R \cdot T) \tag{15}$$

In accordance with Ayers et al. (1980) we neglect any temperature dependence of $\mu_{H_2SO_4,j} - \mu^0_{H_2SO_4}$. This empirically based approach is used in several chemistry transport models to simulate the evaporation of pure sulphuric acid particle in the stratosphere (see e.g. Kokkola et al., 2009; English et. al., 2011 and Hommel et. al., 2011).

We calculate the surface tension and density of the particles comprising a ternary mixture of water, sulphuric acid and ammonium with parameterisations given by Hyvärinen et al. (2005) that combine surface tension parameterisations for (NH$_4$)$_2$SO$_4$–H$_2$O mixtures (Hämeri et al., 2000, Korhonen et al., 1998b), H$_2$SO$_4$–H$_2$O mixtures (Vehkamäki et al., 2002) and NH$_3$–H$_2$O mixtures (King et al. 1930). For the range of conditions in our experiment, where the minimum particle diameter after evaporation is ~50($\pm$10) nm (for experiments 1 and 2). The Kelvin effect only increases the water saturation vapour pressure by maximum value of 1.07 (and the H$_2$SO$_4$ saturation vapour pressure by 1.44, see Supplement Fig. S6) for the particle diameter of 40 nm.

### 3.3.5 Evaporation of H$_2$SO$_4$, SO$_3$ and H$_2$O

We model the gas–particle partitioning (evaporation) of H$_2$SO$_4$ and SO$_3$ using the full moving size distribution method in combination with the Analytic Prediction of Condensation, APC scheme (Jacobson, 2005a). APC is an unconditionally stable numerical discretisation scheme used to solve the condensation equation, Eq. (16). In Eq. (16), we substitute the saturation vapour pressures from Eq. (13) and the measured concentration, $C_{\infty,H_2SO_4(g)}$, (vapour pressure, $p_{\infty,H_2SO_4(g)}$) of H$_2$SO$_{4(g)}$. Based on the motivation given in Sect. 3.3.3 the vapour pressure of SO$_3$, $p_{\infty,SO_3(g)}$, is set to zero.

$$\frac{dm_{i,j}}{dt} = \frac{2 \cdot \pi \cdot (d_i + d_j) \cdot (D_i + D_j) \cdot M_i}{R \cdot T} \cdot \beta_{i,j}(Kn_{i,j}, \alpha_i) \cdot (p_{\infty,i} - p_{s,i,j}) \tag{16}$$

$$\beta_{i,j}(Kn_{i,j}, \alpha_i) = \frac{Kn_{i,j} + 1}{0.377 \cdot Kn_{i,j} + 1 + \dfrac{4}{3 \cdot \alpha_i} \cdot (Kn_{i,j}^2 + Kn_{i,j})}$$

$$Kn_{i,j} = \frac{2 \cdot \lambda_{i,j}}{d_i + d_j}, \qquad \lambda_{i,j} = \frac{3 \cdot (D_i + D_j)}{\sqrt{v_i^2 + v_j^2}} \tag{17}$$

Equation (16) describes the contribution of species $i$ to the mass growth rate of a particle in size bin $j$, $\beta_{i,j}$ is the Fuchs–Sutugin correction factor in the transition region (Fuchs and Sutugin, 1971), $d_i$, $d_j$ correspond to diameters (m) and $D_i$, $D_j$ to diffusion coefficients ($m^2 \cdot s^{-1}$) of the condensing molecule $i$ and the particles in size bin $j$, respectively. $\alpha_i$ is the mass–accommodation coefficient of compounds $i$ and $Kn_{i,j}$ is the non–dimensional Knudsen number, Eq.(17). $\lambda_{i,j}$ is the mean free path (m) and $v_i$, $v_j$ are the thermal speed ($m \cdot s^{-1}$) of the molecule $i$ and the particles in size bin $j$, respectively. Equations (16) and (17) take into account that the condensing molecules have a non–negligible size compared to the size of the smallest particles, and that small particles have non–negligible diffusion coefficients (Lehtinen and Kulmala, 2003).

Based on measurements of $H_2SO_4$ losses in a flow tube reactor, Pöschl et al., (1998) derived a mass accommodation coefficient of $H_2SO_{4(g)}$ on aqueous sulphuric acid, $a_{m,H_2SO_4}$, which was close to unity, with a best fit value of 0.65, a lower limit value of 0.43 and an upper limit of 1.38 (physical limit 1.0). The measured mass accommodation coefficients did not show any dependence on the relative amount of water in the particles (Pöschl et al., 1998). For the model simulations in this work we use unity mass accommodation coefficients. The particle water content is modelled as an equilibrium process with the thermodynamic model (see Sect. 3.3.2).

### 3.3.6 Particle losses

The electric field strength of the stainless–steel CLOUD chamber, in contrast to smog chambers made of Teflon, is very low. Therefore we can neglect electrostatic deposition enhancements (for details on how ADCHAM treats particle wall deposition losses see Roldin et al., 2014). We simulate the particle size–dependent deposition losses with the model from Lai and Nazaroff (2000). The particle deposition loss depends on the friction velocity ($u^*$), which we treat as an unknown model fitting parameter. The best possible agreement between the modelled and measured particle number and volume concentration in the chamber is achieved with a friction velocity of ~$0.2$ $m \cdot s^{-1}$. Thus, for all model results we present in this article we use $u^*=0.2$ $m \cdot s^{-1}$. Dilution losses due to the purified air injected to the CLOUD chamber are also considered in the model.

### 3.3.7 Constraining the thermodynamic properties of sulphate aerosol particles

We use ADCHAM to constrain the values of the thermodynamic equilibrium coefficients, $K_{H_2SO_4}$ and $^xK_{SO_3}$, by treating these coefficients as unknown model fitting parameters. By varying the equilibrium coefficients we search for the best possible agreement (coefficient of determination ($R^2$), see Supplement, Table S1) between the modelled and measured geometric mean diameter (GMD) with respect to particle number. Because experimental results reveal that the sulphate particles did not evaporate completely, they must have been contaminated with a small fraction of effectively non–volatile material (Sect. 3.2).

In the model we address this by assuming that the particles (prior to evaporation) contained either a small fraction of

non–volatile organic material (e.g., secondary organic aerosol, SOA) or that the particles contained small amounts of ammonium, which prevented pure $H_2SO_4$ particle formation and consequently prevented the evaporation. We calculate the initial SOA and ammonium dry particle volume fraction in particle size bin $j$ ($\chi^v_{SOA,j}$ and $\chi^v_{NH4^+,j}$) with Eq. (18) and (19), respectively. Here $d_{SOA}$ and $d_{NH4^+}$ represent an effective particle diameter of SOA and ammonium if all other particle species are removed. For experiment 1 we use $d_{SOA}=60$ nm and $d_{NH4^+}=26$ nm, for experiment 2 $d_{SOA}=43$ nm and $d_{NH4^+}=19$ nm and for experiment 3 $d_{SOA}=38$ nm and $d_{NH4^+}=17$ nm.

$$\chi^v_{SOA,j} = \min\left(\frac{d^3_{SOA}}{d^3_j}, 0.2\right) \tag{18}$$

$$\chi^v_{NH_4^+,j} = \min\left(\frac{d^3_{NH_4^+}}{d^3_j}, 0.05\right) \tag{19}$$

## 4 Results and discussion

In order to fit the modelled particle number size distribution evolution to the observations we performed several hundred simulations where we varied $K_{H_2SO_4}$ and $^xK_{SO_3}$. We summarize these simulations into three main categories (Cases):

1) only $H_2SO_4$ and $H_2O$ evaporation ($^xK_{SO_3}=\infty$), (Case 1)
2) combination of $H_2SO_4$, $H_2O$ and $SO_3$ evaporation, (Case 2) and
3) practically only $SO_3$ and $H_2O$ evaporation, (Case 3).

Case 2 is further divided into two subcategories, Case 2a and 2b. In Case 2a the $H_2SO_4$ is the dominant evaporating S(VI) species while in Case 2b the $SO_3$ is the dominant evaporating S(VI) species.

### 4.1 Particle–phase mole fractions

Figure 2 shows an example of the modelled mole fractions of (**a**) $H_2SO_{4(aq)}$, $\chi_{H_2SO_4}$, and (**b**) $SO_{3(aq)}$, $\chi_{SO_3}$, as a function of the $a_w$ and N:S for Case 2a with equilibrium constants $K_{H_2SO_4}=2.40\cdot10^9$ mol·kg$^{-1}$, and $^xK_{SO_3}=1.43\cdot10^{10}$ at $T=288.8$ K. Fig. 2 reveals that the increase of $\chi_{SO_3}$ as $a_w$ decreases is steeper than for $\chi_{H_2SO_4}$. This is because $H_2SO_{4(aq)}$ formation precedes $SO_3$ formation (see R3). As expected, the highest values of $\chi_{H_2SO_4}$ and $\chi_{SO_3}$ occur when N:S=0 and $a_w$ approaches zero. While N:S increases, $\chi_{H_2SO_4}$ and $\chi_{SO_3}$ decrease gradually and reach lower values when N:S become larger than 0.6.

### 4.2 Particle number size distribution evolution

In Figure 3 we present the particle number size distribution evolution after the shutter of the UV light is closed and the influx of water vapour to the chamber is interrupted for experiment 2, performed at $T=288.8$ K, showing (a) the measured and (b)

the modelled values for Case 2a with $K_{H_2SO_4}=2.40\cdot10^9$ mol·kg$^{-1}$ and $^xK_{SO_3}=1.43\cdot10^{10}$. At the beginning of the evaporation process the particles in the size range from ~60 to ~180 nm in diameter contain approximately 70 mole % $H_2O$; however, this percentage decreases, declining to 15 mole % after 6 h (Fig. 3 (c)). Before $H_2SO_4$ and $SO_3$ start to evaporate from the particles the assumed mole fraction of ammonium is very low (Fig. 3 (d)). However, during the evaporation process $N{:}S$ increases

steadily until it reaches a value of ~0.6 after ~6 h. At this point the particles are ~40 nm in diameter and do not shrink further. This model result is in good agreement with the experimental results reported by Marti et al. (1997) and confirms that $NH_4^+$ effectively stabilizes sulphur particles against evaporation when $N{:}S{\approx}0.6$. Thus, in the stratosphere, even small amounts of a base (such as $NH_3$) can prevent the sulphate particles from shrinking.

## 4.3 Geometric mean diameter shrinkage influenced by relative humidity

Figure 4 compares the measured and modelled GMD evolution as a function of (a) time and (b) RH for experiments 1 and 2 performed at a temperature of $T{=}288.8$ K (Table 1) with $NH_3$ as a particle phase contaminant (see Supplement, Table S1, simulations 1–4 and 13–16 ). The pure liquid saturation vapour pressures of $H_2SO_4$ and $SO_3$ are calculated with Eq. (11) and (12). The model results are in good agreement with the measured GMD trend for Case 1 ($K_{H_2SO_4}=2.00\cdot10^9$ mol·kg$^{-1}$), Case 2a ($K_{H_2SO_4}=2.40\cdot10^9$ mol·kg$^{-1}$ and $^xK_{SO_3}=1.43\cdot10^{10}$), Case 2b ($K_{H_2SO_4}=4.00\cdot10^9$ mol·kg$^{-1}$ and $^xK_{SO_3}=1.54\cdot10^9$) and Case 3

($K_{H_2SO_4}=1.00\cdot10^{11}$ mol·kg$^{-1}$ and $^xK_{SO_3}=3.33\cdot10^7$). The Case 3 simulations give a particle shrinkage that begins somewhat too late and occurs somewhat too rapidly. However, considering the measurement uncertainties it is impossible to constrain the relative contribution of $H_2SO_4$ and $SO_3$ to the observed GMD loss only based on these two experiments (see Sect. 4.4).

With the Aspen Plus Databank pure liquid saturation vapour pressure parameterisations it is also possible to find similarly good agreement between the modelled and observed GMD evolution during experiment 1 and 2 for Cases 1, 2a, 2b

and 3 (Fig. S8) with $NH_3$ as the particle phase contaminant, but with somewhat different values of $K_{H_2SO_4}$ and $^xK_{SO_3}$ (see Supplement, Table S1, simulations 8–11 and 20– 23).

The model simulations with non–volatile and non–water–soluble organics or dimethylamine (DMA) as the particle phase contaminant give nearly identical results as with $NH_3$, both for experiments 1 and 2 (see Supplement Table S1, simulations 6, 7, 17 and 18). In the case of DMA this occurs because it is also a strong enough base to be completely protonated

(all N(–III) is in the form of $NH_4^+$). In the case of an organic contaminant instead of $NH_3$ the model results mainly differ at a later stage of the particle evaporation phase when the $N{:}S$ approaches ~0.5. This is because the evaporation rate does not slow down before all S(VI) is lost when the particles do not contain any base (see Fig. S9). Thus, the modelled GMD shrinkage becomes somewhat faster, when assume organic contamination. Without any particle phase contamination (pure sulphuric acid particles) the particles evaporate faster and completely (see Supplement, Fig. S10).

Instead of explicitly calculating the $H_2SO_4$ activity with the thermodynamic model we derive it directly from the tabulated values of the $H_2SO_4$ chemical potentials as a function of the molality, following Giauque et al. (1960), Eq. (15). With this method we simulate the evaporation of $H_2SO_4$ without explicitly calculating the concentration of $H_2SO_4$ in the particles.

However, since the tabulated chemical potentials from Giauque et al. (1960) are only valid for pure sulphuric acid solutions and temperatures close to 298.15 K it cannot be used if the particle aqueous phase also contains ammonium or other stabilizing molecules.

Based on data from Giauque et al. (1960), Eq. (15) and the pure–liquid saturation vapour pressure parameterization from N–K–L parameterisation (Noppel et al, 2002; Kulmala and Laaksonen, 1990), Eq. (11) the modelled GMD shrinkage is consistent with the observations for experiments 1 and 2, when we consider the Case 1 ($H_2SO_4$ as the only evaporating S(VI) species) and particle phase contamination due to non–volatile non–water–soluble organics (see Supplement, Figure S11 and Table S1, simulations 5, 12, 19 and 24). However, when we use the pure–liquid saturation vapour pressure parameterisation from the Aspen Plus Databank, the modelled particles evaporate earlier (at higher RH) than the observed particles. The reason is that the ASPEN compared to N–K–L parameterisation gives higher saturation vapour pressures (see Supplement, Fig. S5).

### 4.4 Geometric mean diameter shrinkage influenced by relative humidity and temperature

In an attempt to constrain how $K_{H_2SO_4}$ and $^xK_{SO_3}$ depend on the temperature, and the role of $H_2SO_4$ and $SO_3$ on the observed particle diameter shrinkage, as a next step we simulate experiment 3, which expands in temperature. For this experiment the temperature increases gradually from 268 K to 293 K while the absolute humidity remains at a constant value, thus allowing the RH to decrease. Equation (20) describes the modelled temperature dependence of $K_{H_2SO_4}$ and $^xK_{SO_3}$ where the $K_i$ values at $T=288.8$ K ($K_{i,\ 288.8\ K}$) set equal to the values in regard to the model simulations of experiment 1 and 2 (Sect. 4.2):

$$K_i = K_{i,288.8K} \cdot e^{\left(B_i\left(\frac{1}{T} - \frac{1}{288.8}\right)\right)} \tag{20}$$

where $i$ can be either $H_2SO_4$ or $SO_3$. With $B_i=0\ K$ there is no temperature dependence of $K_i$.

For other acids like $HNO_3$, $HCl$ and $HSO_4^-$, $K_i$ decreases with increasing T ($B_i>0$) (Jacobson, 2005a). Que et al. (2011) estimates $B_{H_2SO_4}$ to be 3475 K and $B_{SO_3}$ to be 14245.7 K. Thus, based on this information we would expect the equilibrium reactions R1 and R3 to shift towards the left (more $H_2SO_{4(aq)}$ and $SO_3$ as temperature increases). This would result in a stronger temperature dependence of the $H_2SO_{4(aq)}$ and $SO_3$ saturation vapour pressures over aqueous sulphuric acid droplets (Eq. 13) compared to the temperature dependence expected if we only consider the temperature effect of the pure–liquid saturation vapour pressures (Fig. S5).

Figure 5 compares the measured and modelled GMD evolution during experiment 3. For the simulations we use either the same temperature dependence as suggested by Que et al. (2011) ($B_{H_2SO_4}=3475$ K and $B_{SO_3}=14245.7$ K), or no temperature dependence of $K_{H_2SO_4}$ and $^xK_{SO_3}$ ($B_{H_2SO_4}=0$ K and $B_{SO_3}=0$ K) or weak temperature dependence $B_{H_2SO_4}=0$ K and $B_{SO_3}=-3000$ K. One of these model simulations correspond to Case 1 and the rest to Case 2a (see Supplement, Table S1, simulation 28 and 29, 33, 34 and 36, respectively).

For the Case 1 simulation (see Supplement, Table S1, simulation 28) we use Eq. (15) and the tabulated $H_2SO_4$ chemical potentials from Giauque et al. (1960) to derive the $H_2SO_4$ activity. The particle phase contaminant is assumed to be

non–volatile and non–water–soluble organics. In this simulation the modelled particles grow somewhat too much before they start to shrink. For the Case 2a simulation where the temperature dependences of $K_{H_2SO_4}$ and $^xK_{SO_3}$ are described by the $B_{H_2SO_4}$ and $B_{SO_3}$ values derived by Que *et al.* (2011) (see Supplement, Table S1, simulation 29) the model cannot capture the observed GMD evolution. For the Case 2a simulations with $B_{H_2SO_4}=0$ K and $B_{SO_3}=0$ K (see Supplement, Table S1, simulations 33 and 34) the particle phase contaminant is assumed to be $NH_3$ or non–volatile and non–water–soluble organics. These model simulations, which agree with the observed GMD, indicate that the temperature dependences of $K_{H_2SO_4}$ and $^xK_{SO_3}$ need to be very weak or insignificant ($B_{H_2SO_4}=0$ K and $B_{SO_3}=0$ K). If the particles are contaminated with $NH_3$, $B_{SO_3}$ or $B_{H_2SO_4}$ even needs to be negative for optimum fitting (e.g. $B_{H_2SO_4}=0$ K and $B_{SO_3}=-3000$ K, see Supplement, Table S1, simulations 36). It is also possible to find good agreement between the modelled and measured GMD evolution if one of $B_{H_2SO_4}$ and $B_{SO_3}$ is negative and the other one is positive ($B_{H_2SO_4}=3475$ K and $B_{SO_3}=-10000$ K, see Supplement, Table S1, simulation 31). The $H_2SO_4$ and $SO_3$ pure liquid saturation vapour pressures in these simulations are calculated with Eq. (11) and (12).

If we instead use the pure–liquid saturation vapour pressure parameterizations from the Aspen Plus Databank (which have somewhat weaker temperature dependences than Eq. 11 and 12), the model results captures the observed GMD evolution if both $B_{H_2SO_4}$ and $B_{SO_3}$ are zero and $H_2SO_4$ is the only evaporating (SVI) species (Case 1, see Supplement, Table S1, simulation 50) or the main evaporating S(VI) species (Case 2a, see Supplement, Table S1, simulation 51, see Supplement, Fig. S12).

For Case 2b and 3 simulations in which we assume that $SO_3$ is responsible for most of the S(VI) evaporation, the model can never capture the observed GMD evolution. This is the case regardless of the pure liquid saturation vapour pressure method we use (N–K–L–Nickless or Aspen Plus Databank, see Supplement, Table S1, simulations 42, 48, 52 and 53) .

Based on the simulations of experiment 3 we conclude that most of the S(VI) that evaporated from the particles probably was in the form of $H_2SO_4$ (Cases 1 and 2a). The very weak temperature dependences for $K_{H_2SO_4}$ and $^xK_{SO_3}$ needed for the model to capture the GMD evolution during experiment 3 is surprising and calls for further investigation. Part of the explanation to this could be that the AIOMFAC activity coefficient model is developed based on experimental data derived at 298.15 K. The uncertainty arising from the two different pure liquid saturation vapour pressure parameterisations (temperature dependent) also limits our ability to fully constrain the $K_{H_2SO_4}$ and $^xK_{SO_3}$ values. Based on our experiments and model simulations the equilibrium constant $K_{H_2SO_4}$ should be somewhere in the range $2.0–4.0 \cdot 10^9$ mol·kg$^{-1}$ and the $^xK_{SO_3}$ needs to be larger than $1.4 \cdot 10^{10}$ at a temperature of $288.8 \pm 5$ K. The type of contamination of the sulphate particles ($NH_3$, DMA or a non–volatile non–water–soluble organic compound) does not have a substantial impact on our results and conclusions.

### 4.5 Atmospheric implications

In the following section, we define an effective saturation concentration of $H_2SO_{4(g)}$ $\left(C^*_{H_2SO_4,S}\right)$ as the sum of the saturation concentration of $H_2SO_4$ $\left(C_{H_2SO_4,S}\right)$ and $SO_3$ $\left(C_{SO_3,S}\right)$, based on the assumption of rapid conversion of $SO_{3(g)}$ to

$H_2SO_{4(g)}$, Eq. (21), (see Supplement S5, Fig. S7).

$$C^*_{H_2SO_4,S} = C_{H_2SO_4,S} + C_{SO_3,S} \tag{21}$$

Figure 6 shows the modelled effective $H_2SO_4$ saturation concentration $\left(C^*_{H_2SO_4,S}\right)$ as a function of particle size ($d_p =1\text{–}10^3$ nm) and RH ($0\text{–}100\ \%$). The results are from a model simulation with $K_{H_2SO_4}=2.40\cdot10^9$ mol·kg$^{-1}$ and $^xK_{SO_3}=1.43\cdot10^{10}$, $T=288.8$

K and pure liquid saturation vapour pressures calculated with Eq. (11) and (12). The four different panels (a–d) correspond to simulations using four different values for $N:S$, namely 0, 0.5, 0.75 and 1. In each panel, the contours show the $log_{10}\left(C^*_{H_2SO_4,S}\right)$ levels. For example, the $log_{10}\left(C^*_{H_2SO_4,S}\right)=7$ contour corresponds to an effective $H_2SO_4$ saturation concentration of $10^7$ molecules cm$^{-3}$. These contours provide the $H_2SO_4$ gas–phase concentration at which the net flux of S(VI) to and from the particles is zero (particles neither grow nor shrink).

The observed atmospheric daytime range of the $[H_2SO_{4(g)}]$ is approximately $10^5\text{–}10^8$ molecules cm$^{-3}$, and so we shade this range in Figure 6. When $C^*_{H_2SO_4,S}$ is less than this range (to the upper right in the panel), the particles for most atmospheric daytime conditions will grow by condensation of $H_2SO_4$; when $C^*_{H_2SO_4,S}$ is greater than this (to the lower left in the panel) the particles will for most conditions shrink by evaporation of S(VI); in the shaded range the particles will tend to equilibrate. The larger the mole fraction of bases ($NH_3$) in the aerosol particles the less prone they will be to shrink. When particles are

composed only of S(VI) and $H_2O$ ($N:S=0$) and the concentration of $H_2SO_{4(g)}$ is $10^7$ molecules cm$^{-3}$ all particles smaller than 10 nm will shrink at $RH<13.2\ \%$. For the same $[H_2SO_{4(g)}]$ and $N:S=0.5$ all particles smaller than 10 nm shrink at $RH<12.1\ \%$. However, for $N:S=0.75$ particles smaller than 4 nm shrink at $RH<5.5\ \%$, and if $N:S=1$ only particles smaller than ~1.9 nm shrink, independent of RH except when it is extremely dry ($RH\lesssim1.5\ \%$). With the vapour pressure parameterisations from the Aspen Plus Databank and $K_{H_2SO_4}=4.00\cdot10^9$ and $^xK_{SO_3}=4.55\cdot10^{10}$ the results are almost identical.

These model results demonstrate that sulphuric acid can evaporate from particles or be unable to contribute to their growth for atmospherically relevant conditions, characterized by low relative humility, relatively high temperatures and weak sources of $NH_3$ and $SO_2$. Such environments can be found in the stratosphere and possibly also in the troposphere over large desert regions.

## 5 Summary and conclusions

This study demonstrates, both experimentally and theoretically, the importance of $H_2SO_4$ evaporation from aerosol particles at atmospheric relevant conditions. We measured the sulphate aerosol particle shrinkage below a certain low relative humidity (e.x. $RH\lesssim1.5\%$ for $T=288.8$ K and $RH\lesssim0.7\%$ for $T=268.0$ K) in the CLOUD chamber at CERN. We modelled the sulphur evaporation with ADCHAM. Our model simulation show that:

     i.    the dissociation of $H_2SO_{4(aq)}$ is not complete, and evaporation of $H_2SO_4$ and $H_2O$ can explain the observed particle

30         shrinkage. However, we cannot dismiss the possibility that some of the shrinkage is due to evaporating $SO_3$, which is formed when $H_2SO_{4(aq)}$ is dehydrated.

ii.     the equilibrium rate coefficient for the first dissociation stage of $H_2SO_{4(aq)}$ ($K_{H_2SO_4}$) falls somewhere in the range *2.0–4.0·10⁹* mol·kg⁻¹ at 288.8 ± 5 K.

iii.    the equilibrium coefficient for the dehydration of $H_2SO_4$ ($^xK_{SO_3}$) must at least be larger than $1.4·10^{10}$.

The main factors limiting our estimation of $K_{H_2SO_4}$ are uncertainties in the pure liquid saturation vapour pressure of $H_2SO_4$
and the relative contribution of $SO_3$ to the observed particle evaporation. Other potential sources of error are the uncertainties in the derived activity coefficients, the mass accommodation coefficient of $H_2SO_4$ and solid salt formation during the particle evaporation phase. The model simulations of an experiment where the temperature was gradually increased from 268 to 293 K, indicates that the temperature dependencies of $K_{H_2SO_4}$ and $^xK_{SO_3}$ need to be weak. Future studies should focus on constraining the pure liquid saturation vapour pressures of $H_2SO_4$ and $SO_3$ and the temperature dependence of $K_{H_2SO_4}$ and
$^xK_{SO_3}$.

In order to be able to make an accurate prediction of the sulphate particles influence on global climate, their thermodynamic properties need to be properly described in global climate models. Thus, our constraints on the dissociation, $K_{H_2SO_4}$ and dehydration, $^xK_{SO_3}$ of $H_2SO_4$ are important contributions to the global aerosol-climate model community. The outcome of this study implies that atmospheric modelling studies, especially those dedicated to new particle formation, should
not by default assume that sulphate particles are non-volatile. Models that exclude the evaporation process provide faster particle formation rates which has a misleading effect on the impact of aerosols on climate.

Our results are especially meaningful for high-altitude new particle formation (e.g. in the upper troposphere and stratosphere). It has been previously reported the particle formation (Brock et al., 1995) and ion induced nucleation (Lee et al., 2003; English et al., 2011) as a source of new particles in high altitudes. In the upper troposphere and stratosphere general
circulation models coupled with aerosol dynamics models use aerosol evaporation as a source of [$H_2SO_{4(g)}$] (English et al., 2011). The concentration of $H_2SO_{4(g)}$ drastically affects new particle formation rates. The equilibrium constants for the dissociation and dehydration of $H_2SO_4$ reported in this study are needed to accurately model the sulphate aerosol particle evaporation and concentration of $H_2SO_{4(g)}$. They may also be important to evaluate particle formation schemes (homogeneous, ion-induced) for stratospheric conditions. These schemes are generally constrained based on tropospheric conditions (English
et al., 2011) but applied for stratosphere simulations. Moreover, vapour–phase $H_2SO_4$ in the atmosphere appears to be ubiquitous, even in the absence of photochemistry (Mauldin et al. 2003; Wang et al., 2013); this may partly be due to evaporation of $H_2SO_4$ from aerosol particles.

In a changing climate it will become even more important to understand the thermodynamic properties of the sulphur aerosol particles involved in the development of polar stratospheric clouds and how sulphate aerosols influence the
stratospheric $O_3$ layer. Experiments simulating stratospheric conditions (*T≈200–265* K, *p≈10⁻¹–10⁻³* atm, *RH≥1.0* % and *[$H_2SO_4$]≤10⁸* molec.·cm⁻³), are of great importance. Our results may also assist in explaining the atmospheric sulphur cycle of Venus. The Venusian clouds made up largely of sulphuric acid droplets cover an extended temperature range from 260 K

(upper clouds) to 310 K (middle clouds) and even higher (lower clouds). The scientific understanding of the upper tropospheric and stratospheric sulphate aerosol is of great importance for the global climate and requires further investigation.

**Author Contributions**

G.T. and J.D. designed and performed the experiments. G.T., J.D. and P.R. analysed the data. P.R. developed the model code. P.R. and G.T. performed the simulations. G.T., J. D., L. R., J. T., J. G. S., and A.K. collected the data and contributed to the analysis. G.T, P.R., J.D., and N.M.D assisted in drafting the manuscript. G.T., P. R., J. D., M. B., J. C., R. C. F., M. K., N. M. D., F. S. contributed to scientific interpretation and editing of manuscript. All authors contributed to the development of the CLOUD facility and analysis instruments, and commented on the manuscript.

**Data availability**

Requests for underlying material should be addressed to the corresponding author G.T (george.tsagkogeorgas@tropos.de).

**Acknowledgements**

We would like to thank CERN for supporting CLOUD with important technical and financial resources, and for providing a particle beam from the CERN Proton Synchrotron. We also thank P. Carrie, L.–P. De Menezes, J. Dumollard, R. Guida, K. Ivanova, F. Josa, I. Krasin, R. Kristic, A. Laassiri, O.S. Maksumov, S. Mathot, B. Marichy, H. Martinati, S.V. Mizin, A. Onnela, R. Sitals, H.U. Walther, A. Wasem and M. Wilhelmsson for their important contributions to the experiment. This research has received funding from the EC Seventh Framework Programme (Marie Curie Initial Training Network "CLOUD– ITN" no. 215072 and "CLOUD–TRAIN" no. 316662, ERC–Starting "MOCAPAF" grant no. 57360 and ERC–Advanced "ATMNUCLE" grant no. 227463), the German Federal Ministry of Education and Research (project nos. 01LK0902A and 01LK1222A), the Swiss National Science Foundation (project nos. 200020 135307 and 206620 141278), the Academy of Finland (Center of Excellence project no. 1118615 and other projects 135054, 133872, 251427, 139656, 139995, 137749, 141217, 141451), the Finnish Funding Agency for Technology and Innovation, the Vaisala Foundation, the Nessling Foundation, the Austrian Science Fund (FWF; project no. J3198–N21), the Portuguese Foundation for Science and Technology (project no. CERN/FP/116387/2010), the Swedish Research Council, Vetenskapsradet (grant 2011–5120), the Presidium of the Russian Academy of Sciences and Russian Foundation for Basic Research (grants 08–02–91006–CERN and 12–02– 91522–CERN), the U.S. National Science Foundation (grants AGS1136479, AGS1447056, AGC1439551 and CHE1012293), the PEGASOS project funded by the European Commission under the Framework Program 7 (FP7–ENV–2010–265148), and the Davidow Foundation. We thank the tofTools team for providing tools for mass spectrometry analysis.

P. Roldin would like to thank the Cryosphere–Atmosphere Interactions in a Changing Arctic Climate (CRAICC) and the Swedish Research Council for Environment, Agricultural Sciences and Spatial Planning FORMAS (Project no. 214–2014– 1445) for financial support.

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

**Table 1.** Summary of the experimental conditions: temperature (T), relative humidity (RH), and gaseous sulphuric acid concentration ($[H_2SO_4]_{(g)}$), which is also given as saturation vapour pressure ($p_{sat,H2SO4}$) for each experiment.

| Run No | CLOUD Run No | T (K) | RH (%) | $[H_2SO_4]_{(g)}$, peak (# cm$^{-3}$) | $[H_2SO_4]_{(g)}$, background (# cm$^{-3}$) | $p_{sat,H2SO4}$, peak (Pa) | $p_{sat,H2SO4}$, background (Pa) |
|---|---|---|---|---|---|---|---|
| 1 | 914.01 | 288.8 | 10.1–0.5 | $6.0 \cdot 10^7$ | $1.2 \cdot 10^7$ | $2.3 \cdot 10^{-7}$ | $5.0 \cdot 10^{-8}$ |
| 2 | 914.06 | 288.8 | 3.5–0.5 | $2.3 \cdot 10^8$ | $1.0 \cdot 10^8$ | $9.0 \cdot 10^{-7}$ | $4.2 \cdot 10^{-7}$ |
| 3 | 919.02–04 | 268.0–293.0 | 1.4–0.3 | $1.8 \cdot 10^9$ | $2.0 \cdot 10^8$ | $6.3 \cdot 10^{-6}$ | $2.7 \cdot 10^{-7}$ |

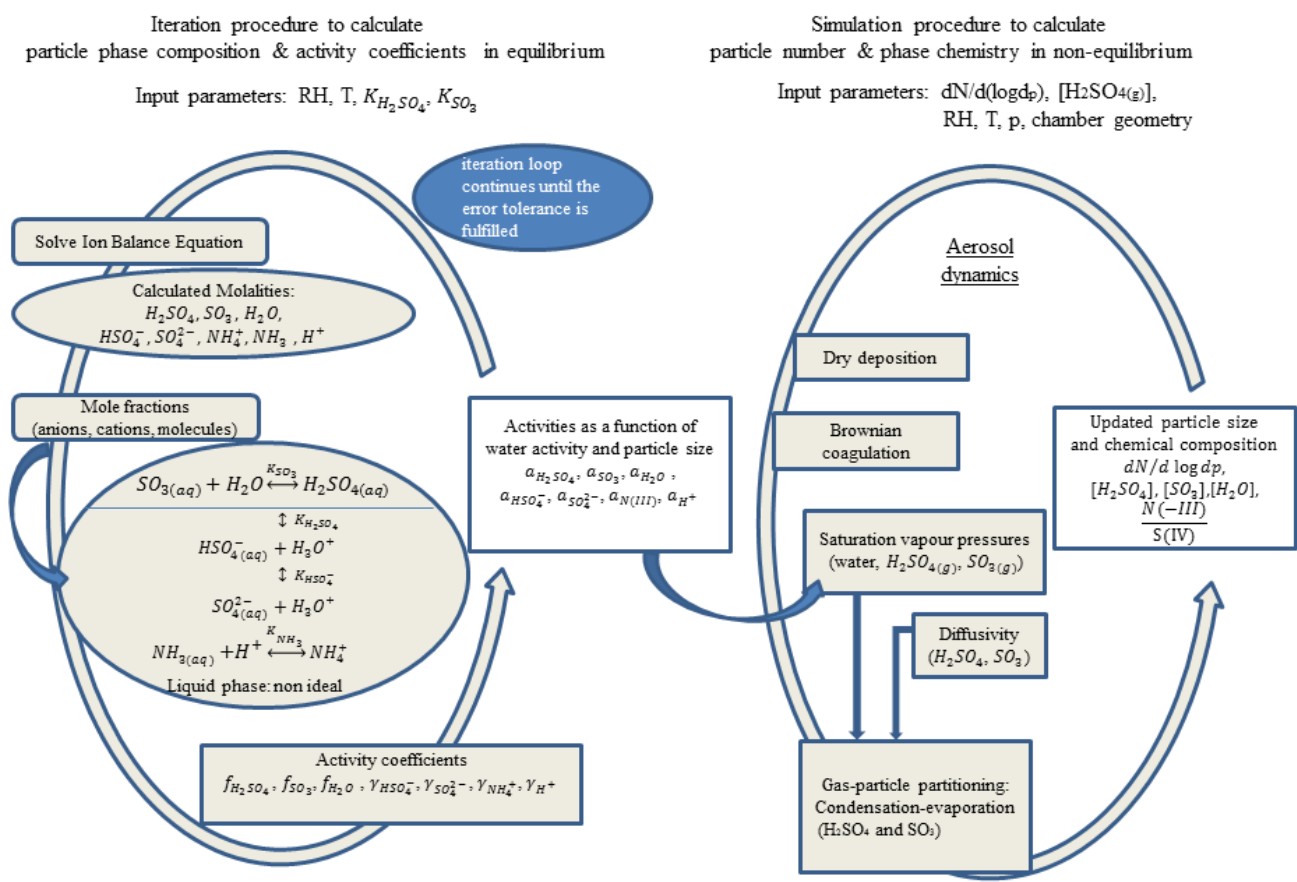

**Figure 1. Schematic of the ADCHAM model optimized for the sulphur particle evaporation at low RH.**

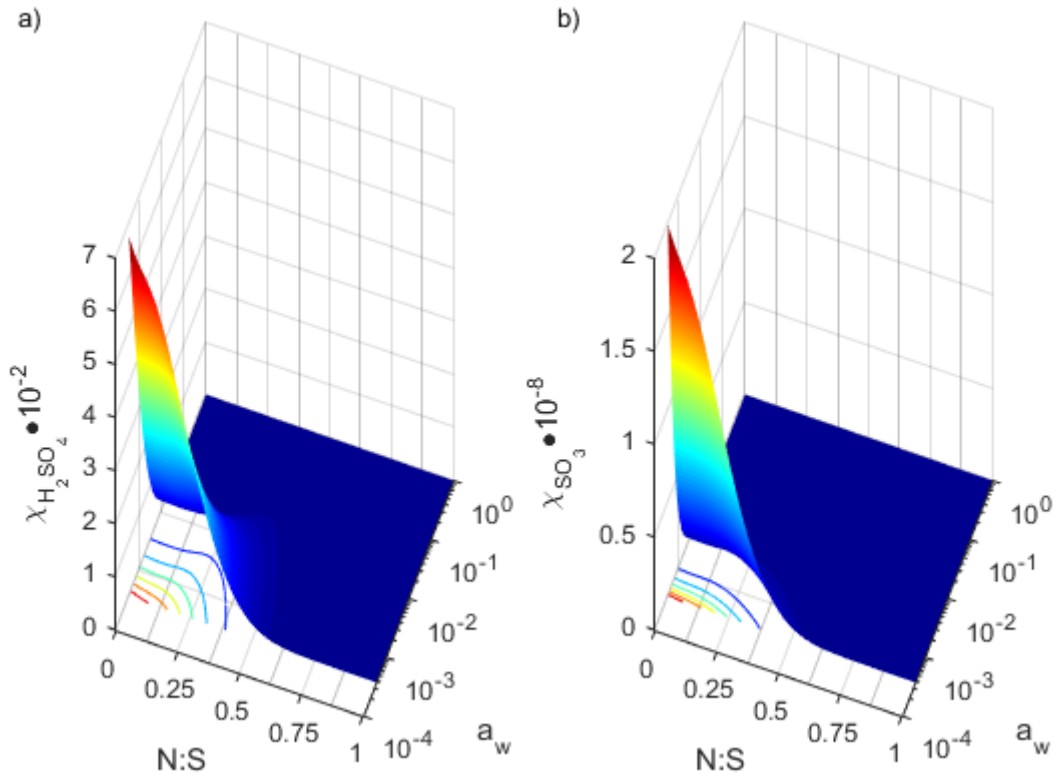

**Figure 2.** Modelled particle–phase mole fractions of (a) $H_2SO_{4(aq)}$, $\chi_{H_2SO_4}$, and (b) $SO_{3(aq)}$, $\chi_{SO_3}$, as a function of the water activity ($a_w$) and the $N{:}S$ for Case 2a which represents the combination of $H_2SO_4$, $H_2O$ and $SO_3$ evaporating species with $H_2SO_4$ being the dominating evaporating S(VI) species. The colour coded contours on x–y axes represent constant particle–phase mole fractions for a) $\chi_{H_2SO_4}=1\text{–}6{\cdot}10^{-2}$ and b) $\chi_{SO_3}=0.3\text{–}1.8{\cdot}10^{-8}$. The equilibrium coefficients are $K_{H_2SO_4}=2.40{\cdot}10^9$ mol·kg$^{-1}$, and $^xK_{SO_3}=1.43{\cdot}10^{10}$ at $T=288.8$ K.

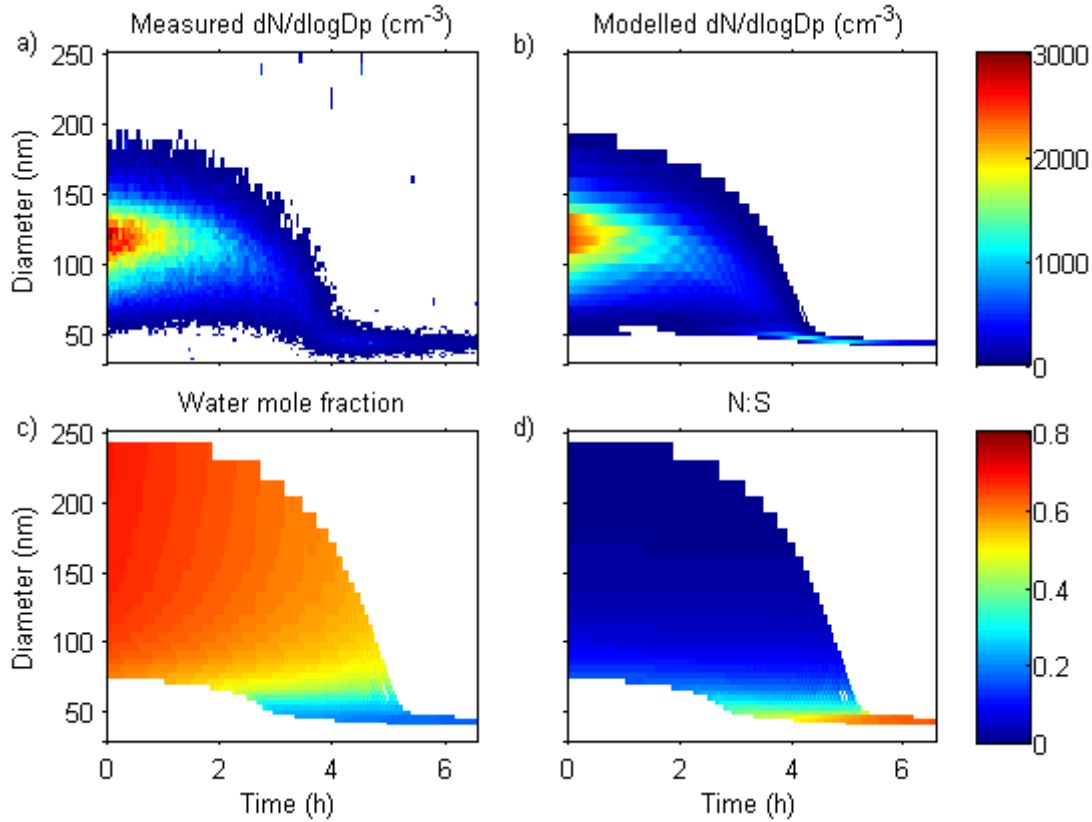

**Figure 3. Particle shrinkage at low RH.** Measured (a) and modelled (b) particle number size distribution evolution during experiment 2 performed at $T=288.8$ K for Case 2a with $H_2SO_4$ being the dominating evaporating S(VI) species, $K_{H_2SO_4}=2.40\cdot10^9$ mol·kg$^{-1}$ and $^xK_{SO_3}=1.43\cdot10^{10}$. Figures (c) and (d) show the modelled particle water mole fraction, $\chi_{H_2O}$ and *N:S*, respectively.

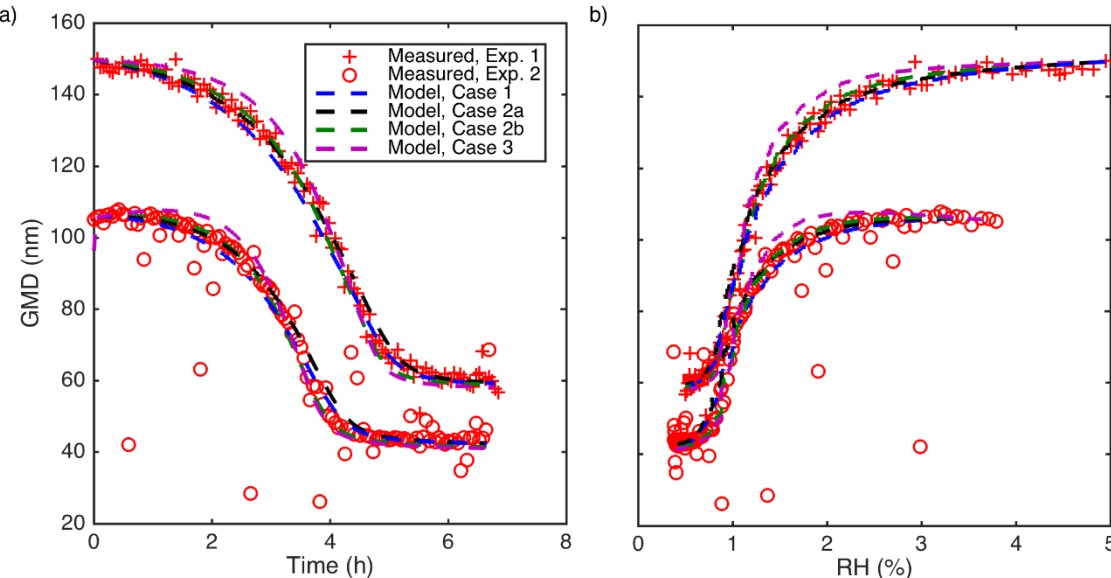

**Figure 4. Measured and modelled GMD evolution as a function of (a) time and (b) RH for experiments 1 and 2 performed at *T=288.8* K. The modelled particles are composed of S(VI), H₂O and NH₃ as a particle phase contaminant. The simulations correspond to Case 1 with H₂SO₄ being the only evaporating S(VI) species, $K_{H_2SO_4}$=2.00·10⁹ mol·kg⁻¹, Case 2a with H₂SO₄ being the dominating evaporating S(VI) species, $K_{H_2SO_4}$=2.40·10⁹ mol·kg⁻¹ and $^xK_{SO_3}$ =1.43·10¹⁰, Case 2b with SO₃ being the dominating evaporating S(VI) species, $K_{H_2SO_4}$ =4.00·10⁹ mol·kg⁻¹ and $^xK_{SO_3}$ =1.54·10⁹ and Case 3 with SO₃ being the only evaporating S(VI) species, $K_{H_2SO_4}$=1.00·10¹¹ mol·kg⁻¹ and $^xK_{SO_3}$=3.33·10⁷ (see Supplement, Table S1, simulations 1–4 and 13–16). The pure liquid saturation vapour pressures of H₂SO₄ and SO₃ are calculated with Eq. (11), N–K–L parameterisation, (Kulmala and Laaksonen (1990) and Noppel et al., 2002) and Eq. (12) (Nickless, 1968), respectively.**

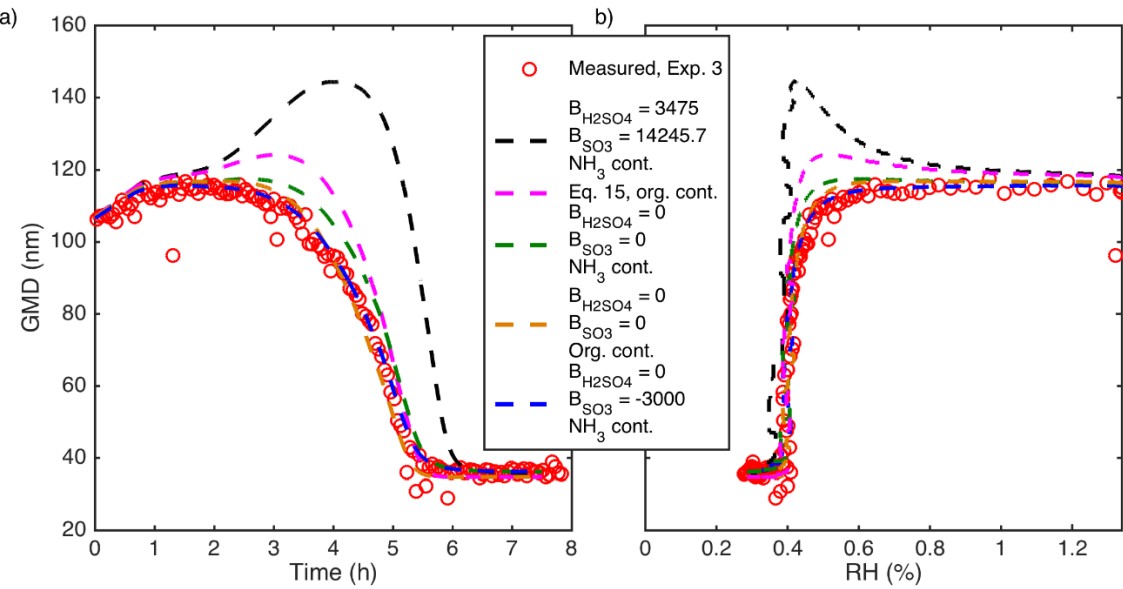

**Figure 5. Measured and modelled GMD evolution as a function of (a) time and (b) RH for experiment 3 performed at a temperature range from 268 K to 293 K. The modelled particles are composed of S(VI), $H_2O$ and either NH3 or non–volatile, non–water–soluble organics as a particle phase contaminant. The simulations correspond to Case 1 (the $H_2SO_4$ activity is calculated with use of Eq. (15) and the tabulated $H_2SO_4$ chemical potentials from Giauque et al. (1960), see Supplement, Table S1, simulation 28) and Case 2a, $K_{H_2SO_4}=2.40\cdot10^9$ mol·kg$^{-1}$ and $^xK_{SO_3}=1.43\cdot10^{10}$ at $T=288.8$ K, (see Supplement, Table S1, simulations 29, 33, 34 and 36). The pure liquid saturation vapour pressures of $H_2SO_4$ and $SO_3$ are calculated with Eq. (11) (Kulmala and Laaksonen (1990) and Noppel et al., 2002) and Eq. (12) (Nickless, 1968), respectively.**

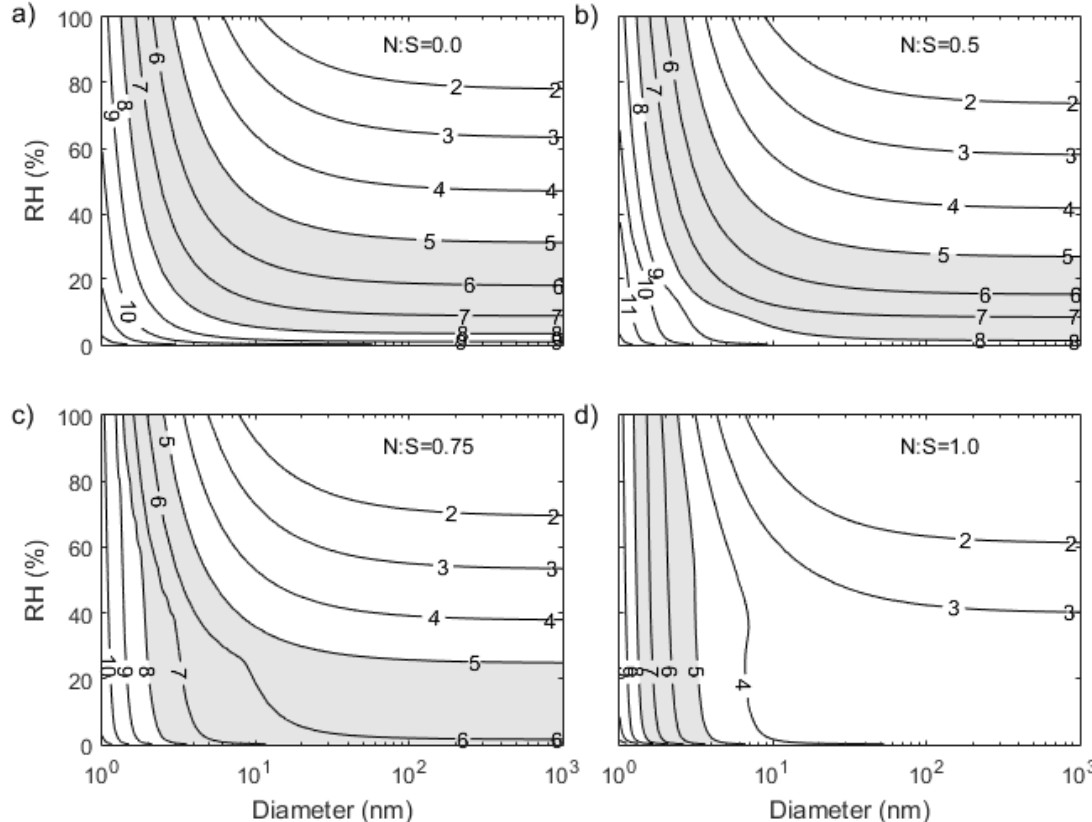

**Figure 6. Modelled effective H₂SO₄ saturation concentration, $C^*_{H_2SO_4,S}$, (molecules·cm⁻³), expressed in $log_{10}(C^*_{H_2SO_4,S})$, at *T=288.8* K, *RH 0–100 %* and particle diameters in the range from 1 to 10³ nm. The contours represent H₂SO₄ gas–phase concentrations, e.g. $log_{10}(C^*_{H_2SO_4,S})$, =7 corresponds to $C^*_{H_2SO_4,S}=10^7$ *molecules·cm⁻³*. The grey shading indicates the atmospheric range of H₂SO₄ (*10⁵⁻ 10⁸ cm⁻³*). The results correspond to particles composed (a) only of S(VI) and H₂O (*N:S=0*), (b) with *N:S=0.5*, (c) with *N:S=0.75* and (d) with *N:S=1*. The equilibrium constants are $K_{H_2SO_4}=2.40·10^9$ mol·kg⁻¹ and $^xK_{SO_3}=1.43·10^{10}$. The pure liquid saturation vapour pressures of H₂SO₄ and SO₃ are calculated with Eq. (11) and (12).**