# Peer review of "Evaporation of sulphate aerosols at low relative humidity"

_Atmospheric Chemistry and Physics, 2016_

## Referee Comment (RC1) · Anonymous Referee #1 · 1 Mar 2017

In general, this was a well-designed study and the results are of sufficient importance and novelty to merit publication in Atmospheric Chemistry and Physics. Although the conclusions are not particularly precise or substantial, the approach to the problem is commendable and the constraints on the dissociation and dehydration equilibrium constants will likely be useful to the modeling community. It is clear that the existing literature has been well reviewed and that the current work contributes to this body of literature. A major issue with the manuscript, however, is that it is too long and thus a bit cumbersome and inaccessible. For example, the introductory material is 4 pages while the methods section is 9 pages. Overall, there are 19 pages of text, a very large table and 8 multi-paneled figures. It is the reviewer's opinion that the manuscript would benefit greatly from a reduction in length and more succinct presentation of the work.

---

## Referee Comment (RC2) · Anonymous Referee #2 · 6 Mar 2017

This is a nice, if lengthy, manuscript on sulfuric acid at low RH that will be of interest to the ACP community. It is well written and appears to be soundly executed work.

Couple of comments for improvement in regards to equations 2-5:

The reason for the change in the basis of the activity coefficients in equations 2,5 vs equation 4 is not clear. A reason should be given. For clarity, the symbols for the activity coefficients should be different if a different basis is used. Please correct for eq 4 and 12.

There are also a few minor typos, including:

(4) Abstract (... and then measuring evaporation...) should be (... and then measured evaporation...). (5) Throughout, pick either sulfuric or sulphuric. (6) 'gases', not

'gasses'; 'nucleus', not nucleous; 'model' instead of 'module'?

---

## Author Response (AR1)

**Response to Referees' comments and a marked-up revised submitted manuscript**

Dear Co-editor and Referees,

Below we attempt to reply thoroughly to Referees comments and we also state the main revisions of the manuscript. We also provide a marked-up revised manuscript showing all the changes made compared to the initial version of our manuscript. All the changes in the revised manuscript are coloured in red instead of using track changes in Word, which complicates the easy process of reading.

**Referees' comments (in blue) and author's response (in red)**

**Referee #1**

In general, this was a well-designed study and the results are of sufficient importance and novelty to merit publication in Atmospheric Chemistry and Physics.

We thank the reviewer for the encouraging words.

Although the conclusions are not particularly precise or substantial, the approach to the problem is commendable and the constraints on the dissociation and dehydration equilibrium constants will likely be useful to the modelling community.

We agree. Thus, we reform the summary and conclusions section. We also discuss the importance to include in models accurate descriptions of the sulphate aerosol evaporation process in the stratosphere, and to consider the volatility of sulphate particles in new particle formation studies.

It is clear that the existing literature has been well reviewed and that the current work contributes to this body of literature.

We would like to thank the reviewer and to mention that we put our efforts to connect parts of the atmospheric climate puzzle through a precise experimental and modelling study.

A major issue with the manuscript, however, is that it is too long and thus a bit cumbersome and inaccessible. For example, the introductory material is 4 pages while the methods section is 9 pages. Overall, there are 19 pages of text, a very large table and 8 multi-paneled figures. It is the reviewer's opinion that the manuscript would benefit greatly from a reduction in length and more succinct presentation of the work.

We consider carefully this remark about the manuscripts' length. We reduce an extensive amount of the introductory material and also the length of the methods section where it is appropriate. In regard to the description of the modelling work, included in this section, we consider the information valuable and it is difficult to proceed with major changes because it will limit the comprehension and reproducibility of the study. Additionally, we reduce the number of figures from the manuscript and attach them to the supplement. We also attach the very large table to the supplement.

**Referee #2**

This is a nice, if lengthy, manuscript on sulfuric acid at low RH that will be of interest to the ACP community. It is well written and appears to be soundly executed work.

We would like to thank the reviewer for his supporting comments. We take care of the manuscripts' length and reduce
5     it where we consider it appropriate (see response comment to Referee #1).

Couple of comments for improvement in regards to equations 2-5: The reason for the change in the basis of the activity coefficients in equations 2,5 vs equation 4 is not clear. A reason should be given. For clarity, the symbols for the activity coefficients should be different if a different basis is used. Please correct for eq 4 and 12.

The simple explanation why to use the mole fraction based form of Eq. 4 in the model due to how we thought it was
10    easiest to implement Eq. 4 in ADCHAM, and how we did it. The eNRTL model that we use provide mole fraction based activity coefficients. Thus, instead of converting them to molality based activity coefficients, we expressed Eq. 4 in the mole fraction based form. Moreover, for a hypothetical very low water activity ($a_w$) a non-negligible fraction of the solvents will be $H_2SO_4$ and $SO_3$ and then we believe that it is questionable to express the species concentrations using molalities, where water is assumed to be the only solvent. On page 11, line 6 we suggest that we add the following clarifying text:

15    "The Eq. 4 is given in a mole fraction based form for the following reasons: a) the eNRTL provides mole fraction based activity coefficients, and b) if Eq. 4 would be applied for $a_w$ that are even lower than considered in this work, the assumption of using molalities, i.e. where water is considered to be the only solvent, will not be acceptable."

We now correct the notation of the mole fraction based activity coefficients in Eq. 4 and 12. Explicitly, we now use $f_{H_2O}$, $f_{H_2SO_4}$ and $f_{SO_3}$ instead of $\gamma_{H_2O}$, $\gamma_{H_2SO_4}$ and $\gamma_{SO_3}$. We change this notation on the manuscript in page 9, Fig. 1, Eq. 4,
20    Eq. 12, and in the supplement Fig. S3, Eq. S1 and Eq. S2.

There are also a few minor typos, including: (4) Abstract (. . . and then measuring evaporation. . .) should be (. . . and then measured evaporation. . .). (5) Throughout, pick either sulfuric or sulphuric. (6) 'gases', not 'gasses'; 'nucleus', not nucleous; 'model' instead of 'module'?

We would like to thank the reviewer for his comment pointing on minor typos. We change accordingly the text.
25    In general, the main changes in the revised manuscript are:

- Reduction in the number of plots. Figures and corresponding text which are removed from the manuscript are now attached to the supplement.

- Remove the very large table and attach it to the supplement.

- Reformulation of the abstract and sections so as to increase the clarity and/or reduce the length.

30    - Explanation for change in the basis of activity coefficients and replace symbol $\gamma$ with symbol $f$ (representing mole fraction based activity coefficients) where is necessary (ex. in text, Fig. 1, Eq. 4, Eq. 12, Fig. S3, Eq. S1 and Eq. S2).

- Additional comments in section 4.5 and 5 relevant to the atmospheric importance of our experimental and modelling work (e.x. on the volatility of sulphate particles, on the accurate description of the thermodynamic properties of sulphate aerosol in stratosphere).
- Add a number to the first equation, the kappa equation. Thus, all the following equation numbers now shift + 1 unit.
- Typos.

[revised manuscript text omitted]

5    **$K_{H_2SO_4}$ and $^xK_{SO_3}$, the assumed species composition of the particle contamination (Con.), and the source to the pure–liquid saturation vapour pressure parameterizations are given.**

| Exp. No. | Sim. No | Case | $B_{H_2SO_4}$ (K) | $B_{SO_3}$ (K) | $K_{H_2SO_4,288K}$ (mol·kg$^{-1}$) | $^xK_{SO_3,288K}$ | Con. | Vap. pres. | $R^2$ |
|---|---|---|---|---|---|---|---|---|---|
| 1 | 1 | [a]1 | 0 | 0 | $2.00\cdot10^9$ | $\infty$ | NH$_3$ | N–K–L, Nickless | 0.994 |
| 1 | 2 | [b]2a | 0 | 0 | $2.40\cdot10^9$ | $1.43\cdot10^{10}$ | NH$_3$ | N–K–L, Nickless | 0.994 |
| 1 | 3 | [c]2b | 0 | 0 | $4.00\cdot10^9$ | $1.54\cdot10^9$ | NH$_3$ | N–K–L, Nickless | 0.996 |
| 1 | 4 | [d]3 | 0 | 0 | $1.00\cdot10^{11}$ | $3.33\cdot10^7$ | NH$_3$ | N–K–L, Nickless | 0.992 |
| 1 | 5 | 1 | 0 | 0 | ** | $\infty$ | Org. | N–K–L, Nickless | 0.992 |
| 1 | 6 | 2a | 0 | 0 | $2.40\cdot10^9$ | $1.43\cdot10^{10}$ | Org. | N–K–L, Nickless | 0.995 |
| 1 | 7 | 2a | 0 | 0 | $2.40\cdot10^9$ | $1.43\cdot10^{10}$ | DMA | N–K–L, Nickless | 0.993 |
| 1 | 8 | 1 | 0 | 0 | $3.80\cdot10^9$ | $\infty$ | NH$_3$ | ASPEN | 0.990 |
| 1 | 9 | 2a | 0 | 0 | $4.00\cdot10^9$ | $4.55\cdot10^{10}$ | NH$_3$ | ASPEN | 0.993 |
| 1 | 10 | 2b | 0 | 0 | $5.00\cdot10^9$ | $5.00\cdot10^9$ | NH$_3$ | ASPEN | 0.995 |
| 1 | 11 | 3 | 0 | 0 | $1.00\cdot10^{11}$ | $5.00\cdot10^7$ | NH$_3$ | ASPEN | 0.990 |
| 1 | 12 | 1 | 0 | 0 | ** | $\infty$ | Org. | ASPEN | 0.888 |
| 2 | 13 | 1 | 0 | 0 | $2.00\cdot10^9$ | $\infty$ | NH$_3$ | N–K–L, Nickless | 0.870 |
| 2 | 14 | 2a | 0 | 0 | $2.40\cdot10^9$ | $1.43\cdot10^{10}$ | NH$_3$ | N–K–L, Nickless | 0.869 |
| 2 | 15 | 2b | 0 | 0 | $4.00\cdot10^9$ | $1.54\cdot10^9$ | NH$_3$ | N–K–L, Nickless | 0.871 |
| 2 | 16 | 3 | 0 | 0 | $1.00\cdot10^{11}$ | $3.33\cdot10^7$ | NH$_3$ | N–K–L, Nickless | 0.868 |
| 2 | 17 | 2a | 0 | 0 | $2.40\cdot10^9$ | $1.43\cdot10^{10}$ | Org. | N–K–L, Nickless | 0.870 |
| 2 | 18 | 2a | 0 | 0 | $2.40\cdot10^9$ | $1.43\cdot10^{10}$ | DMA | N–K–L, Nickless | 0.869 |
| 2 | 19 | 1 | 0 | 0 | ** | $\infty$ | Org. | N–K–L, Nickless | 0.868 |
| 2 | 20 | 1 | 0 | 0 | $3.80\cdot10^9$ | $\infty$ | NH$_3$ | ASPEN | 0.867 |
| 2 | 21 | 2a | 0 | 0 | $4.00\cdot10^9$ | $4.55\cdot10^{10}$ | NH$_3$ | ASPEN | 0.870 |
| 2 | 22 | 2b | 0 | 0 | $5.00\cdot10^9$ | $5.00\cdot10^9$ | NH$_3$ | ASPEN | 0.871 |
| 2 | 23 | 3 | 0 | 0 | $1.00\cdot10^{11}$ | $5.00\cdot10^7$ | NH$_3$ | ASPEN | 0.867 |
| 2 | 24 | 1 | 0 | 0 | ** | $\infty$ | Org. | ASPEN | 0.510 |
| 3 | 25 | 1 | 0 | 0 | $2.00\cdot10^9$ | $\infty$ | NH$_3$ | N–K–L, Nickless | 0.841 |

| | | | | | | | | | |
|---|---|---|---|---|---|---|---|---|---|
| 3 | 26 | 1 | 0 | 0 | $2.00 \cdot 10^9$ | $\infty$ | Org. | N–K–L, Nickless | 0.905 |
| 3 | 27 | 1 | 3475* | 0 | $2.00 \cdot 10^9$ | $\infty$ | $NH_3$ | N–K–L, Nickless | 0.534 |
| 3 | 28 | 1 | 0 | 0 | ** | $\infty$ | Org. | N–K–L, Nickless | 0.967 |
| 3 | 29 | 2a | 3475* | 14245.7* | $2.40 \cdot 10^9$ | $1.43 \cdot 10^{10}$ | $NH_3$ | N–K–L, Nickless | 0.611 |
| 3 | 30 | 2a | 3475* | 0 | $2.40 \cdot 10^9$ | $1.43 \cdot 10^{10}$ | $NH_3$ | N–K–L, Nickless | 0.825 |
| 3 | 31 | 2a | 3475* | −10000 | $2.40 \cdot 10^9$ | $1.43 \cdot 10^{10}$ | $NH_3$ | N–K–L, Nickless | 0.992 |
| 3 | 32 | 2a | 0 | 14245.7* | $2.40 \cdot 10^9$ | $1.43 \cdot 10^{10}$ | $NH_3$ | N–K–L, Nickless | 0.839 |
| 3 | 33 | 2a | 0 | 0 | $2.40 \cdot 10^9$ | $1.43 \cdot 10^{10}$ | $NH_3$ | N–K–L, Nickless | 0.981 |
| 3 | 34 | 2a | 0 | 0 | $2.40 \cdot 10^9$ | $1.43 \cdot 10^{10}$ | Org. | N–K–L, Nickless | 0.991 |
| 3 | 35 | 2a | 0 | −10000 | $2.40 \cdot 10^9$ | $1.43 \cdot 10^{10}$ | $NH_3$ | N–K–L, Nickless | 0.860 |
| 3 | 36 | 2a | 0 | −3000 | $2.40 \cdot 10^9$ | $1.43 \cdot 10^{10}$ | $NH_3$ | N–K–L, Nickless | 0.993 |
| 3 | 37 | 2b | 3475* | 14245.7* | $4.00 \cdot 10^9$ | $1.54 \cdot 10^9$ | $NH_3$ | N–K–L, Nickless | 0.937 |
| 3 | 38 | 2b | 3475* | 0 | $4.00 \cdot 10^9$ | $1.54 \cdot 10^9$ | $NH_3$ | N–K–L, Nickless | 0.819 |
| 3 | 39 | 2b | 3475* | − 10000 | $4.00 \cdot 10^9$ | $1.54 \cdot 10^9$ | $NH_3$ | N–K–L, Nickless | 0.458 |
| 3 | 40 | 2b | 3475* | 5000 | $4.00 \cdot 10^9$ | $1.54 \cdot 10^9$ | $NH_3$ | N–K–L, Nickless | 0.918 |
| 3 | 41 | 2b | 0 | 14245.7* | $4.00 \cdot 10^9$ | $1.54 \cdot 10^9$ | $NH_3$ | N–K–L, Nickless | 0.953 |
| 3 | 42 | 2b | 0 | 0 | $4.00 \cdot 10^9$ | $1.54 \cdot 10^9$ | $NH_3$ | N–K–L, Nickless | 0.685 |
| 3 | 43 | 2b | 0 | − 10000 | $4.00 \cdot 10^9$ | $1.54 \cdot 10^9$ | $NH_3$ | N–K–L, Nickless | 0.260 |
| 3 | 44 | 3 | 3475* | 14245.7* | $1.00 \cdot 10^{11}$ | $3.33 \cdot 10^7$ | $NH_3$ | N–K–L, Nickless | 0.903 |
| 3 | 45 | 3 | 3475* | 0 | $1.00 \cdot 10^{11}$ | $3.33 \cdot 10^7$ | $NH_3$ | N–K–L, Nickless | 0.571 |
| 3 | 46 | 3 | 3475* | − 10000 | $1.00 \cdot 10^{11}$ | $3.33 \cdot 10^7$ | $NH_3$ | N–K–L, Nickless | 0.146 |
| 3 | 47 | 3 | 0 | 14245.7* | $1.00 \cdot 10^{11}$ | $3.33 \cdot 10^7$ | $NH_3$ | N–K–L, Nickless | 0.898 |
| 3 | 48 | 3 | 0 | 0 | $1.00 \cdot 10^{11}$ | $3.33 \cdot 10^7$ | $NH_3$ | N–K–L, Nickless | 0.420 |
| 3 | 49 | 3 | 0 | − 10000 | $1.00 \cdot 10^{11}$ | $3.33 \cdot 10^7$ | $NH_3$ | N–K–L, Nickless | 0.138 |
| 3 | 50 | 1 | 0 | 0 | $3.80 \cdot 10^9$ | $\infty$ | $NH_3$ | ASPEN | 0.991 |
| 3 | 51 | 2a | 0 | 0 | $4.00 \cdot 10^9$ | $4.55 \cdot 10^{10}$ | $NH_3$ | ASPEN | 0.992 |
| 3 | 52 | 2b | 0 | 0 | $5.00 \cdot 10^9$ | $5.00 \cdot 10^9$ | $NH_3$ | ASPEN | 0.880 |
| 3 | 53 | 3 | 0 | 0 | $1.00 \cdot 10^{11}$ | $5.00 \cdot 10^7$ | $NH_3$ | ASPEN | 0.540 |

\* Values from Que et al. (2011).

** Simulation with the $H_2SO_4$ activity derived from Eq. (15) using the thermodynamic data from Giauque et al. (1960)

[a] Case 1: Only evaporation of $H_2SO_4$.

[b] Case 2a: Both $H_2SO_4$ and $SO_3$ evaporate from the particles. $H_2SO_4$ is the main evaporating species at $T=288.8$ K.

[c] Case 2b: Both $H_2SO_4$ and $SO_3$ evaporate from the particles. $SO_3$ is the main evaporating species at $T=288.8$ K.

5      [d] Case 3: $SO_3$ is completely dominating the evaporation.

**S1 AMS measurements**

The evaporation of sulphate particles based on AMS measurements (Fig. S1 (a)) showed that the particles were composed almost exclusively of sulphuric acid. Calculations of the kappa value $\kappa$, based on the AMS measurements, yield a value close to the $\kappa$ for pure sulphuric acid particles (see Fig. S2).

[Figure]

**Figure S1. (a) Sulphate mass size distribution ug·m$^{-3}$ (from AMS data) and (b) gas–phase H$_2$SO$_4$ concentration (from CIMS data) increases until reaches a peak value during the aerosol particle evaporation experiment 2 performed at *T=288.8* K.**

[Figure]

**Figure S2.** Hygroscopicity kappa ($\kappa$), based on the AMS measurements, of mixed particles as a function of time for experiment 3. $\kappa$ derived from the hygroscopicities of the components (assumed the lower and higher $\kappa$ values for bases like ammonium sulphate, $\kappa_{(NH_4)_2SO_4}$=0.47 and ammonium bisulfate, $\kappa_{NH_4HSO_4}$=0.56 (Topping et al., 2005; Petters and Kreidenweis 2007), and organics with *O:C*=0, $\kappa_{Org}$=0.0 and *O:C*=1, $\kappa_{Org}$=0.3 (Massoli et al., 2010)) and their respective volume fractions by applying the Zdanovskii–Stokes–Robinson (ZSR) mixing rule. For the calculation of the volume concentration of each compound assumed liquid phase density of SO₄, NH₄, NO₃, Chl, Org constituents (http://cires1.colorado.edu/jimenez–group/wiki). The difference in percentage of $\kappa$ values calculated for the two extreme cases of $\kappa_{(NH_4)_2SO_4}$=0.47, $\kappa_{NH_4HSO_4}$=0.56 is 0.4 %, while for $\kappa_{Org}$=0.0 and $\kappa_{Org}$=0.3 is 1 %. The result shows a $\kappa$ very close to that of pure sulphuric acid (Sullivan et al., 2010).

**S2 Mole fraction based activity coefficients of H₂SO₄ and SO₃ and water activity**

[Figure]

**Figure S3. Modelled mole fraction based activity coefficient of (a) H₂SO₄ ($f_{H_2SO_4}$) with equilibrium constant $K_{H_2SO_4}$=2.40·10⁹ mol·kg⁻¹, and (b) SO₃ ($f_{SO_3}$) with equilibrium constant $^xK_{SO_3}$=1.43·10¹⁰, at T=288.8 K, as a function of the water activity, $a_w$, on the y–axis and N:S on the x–axis. The colour coded contours on x–y axes represent constant activity coefficient for a) $f_{H_2SO_4}$=0.8–2.2 and b) $f_{SO_3}$=0.8–1.8.**

[Figure]

[Figure]

**Figure S4. (a) Modelled water activity curves and b) degree of dissociation of HSO₄⁻ as a function of water mass fraction in aqueous solutions of H₂SO₄ and mixtures of (NH₄)₂SO₄ and H₂SO₄. The model simulations and measurements were performed at 298 K. The modelled water activity curves are lines colour coded. The purple curve corresponds to pure sulphuric acid, blue and cyan curves to 1:2 and 1:1 molar ratio of *(NH₄)₂SO₄:H₂SO₄* and red curve to pure ammonium sulphate. The measured water activity curve is symbol coded. The purple circle symbol corresponds to H₂SO₄₍aq₎ (Staples 1981). (b) the modelled degree of dissociation, $a_{HSO_4^-}$, curves are lines colour coded (corresponding to same aqueous solutions as the curves in Fig. (S4 (a)). The measured degree of dissociation is symbol colour coded (purple squares corresponds to H₂SO₄₍aq₎, Myhre et al. (2003), cyan triangles to the 1:1 *(NH₄)₂SO₄:H₂SO₄* mixture, Dawson et al. (1986)). The model results can be compared with analogous results in Fig. 10 from Zuend et al., 2011.**

**S3 Saturation vapour pressure parameterizations**

[Figure]

**Figure S5. Saturation vapour pressures for $H_2SO_4$ and $SO_3$. Comparison among two different pure liquid saturation vapour pressure parameterizations (a) for $H_2SO_4$ and (b) for $SO_3$. In panel (a) the blue curve corresponds to the parameterization from the work of Kulmala and Laaksonen (1990), which was optimized by Noppel et al., 2002 (N–K–L parameterization), Eq. (11). The black curve corresponds to the parameterization from Que et al., 2011 (original Aspen Plus Databank). In panel (b) the blue curve corresponds to the parameterization from the work of Nickless (1968), Eq. (12) and the black curve to the parameterization from Que et al., 2011 (original Aspen Plus Databank).**

**S4 The Kelvin effect**

[Figure]

Figure S6. The Kelvin effect for experiment 2 at $T$=288.8 K for Case 2a ($K_{H_2SO_4}$=2.40·10$^9$ mol·kg$^{-1}$ and $^x K_{SO_3}$=1.43·10$^{10}$) illustrates the increase in (a) water (white contours correspond to the Kelvin terms $S_{Kelvin,H_2O}$=1.02–1.38) and (b) H$_2$SO$_4$ (white contours represent the Kelvin terms $S_{Kelvin,H_2SO_4}$=1.2–6.0) saturation vapour pressures. The minimum particle size diameter for experiment 2 is ~40 nm, so the maximum value of the Kelvin term is ~1.44 for sulphuric acid.

**S5 Saturation concentration of H₂SO₄ and SO₃**

We can calculate the saturation concentration of $H_2SO_4$ ($C_{H_2SO_4,S}$, Eq. S1) and $SO_3$ ($C_{SO_3,S}$, Eq. S2) in µg·m$^{-3}$ (Fig. S7) with the $H_2SO_4$ dissociation equilibrium coefficients, $K_{H_2SO_4}=2.4\cdot10^9$ mol·kg$^{-1}$, and $^xK_{SO_3}=1.43\cdot10^{10}$, based on the mole fractions (Fig. 2), the modelled mole fraction based activity coefficients (Fig. S3), the pure liquid saturation vapours pressure parameterizations, Eq. (11) and (12), and the Kelvin effect, Eq. (14).

$$C_{H_2SO_4,S} = \frac{p_{0,H_2SO_4} \cdot x_{H_2SO_4} \cdot f_{H_2SO_4} \cdot C_{k,H_2SO_4}}{R \cdot T \cdot M_{H_2SO_4}} \tag{S1}$$

$$C_{SO_3,S} = \frac{p_{0,SO_3} \cdot x_{SO_3} \cdot f_{SO_3} \cdot C_{k,SO_3}}{R \cdot T \cdot M_{SO_3}} \tag{S2}$$

For almost dry conditions ($a_w=3.7\cdot10^{-4}$) and N:S=0, $C_{H_2SO_4,S}\approx2.6$ µg·m$^{-3}$ and $C_{SO_3,S}\approx8.8$ µg·m$^{-3}$. However, as long as $a_w$ is larger than $1.3\cdot10^{-3}$, $C_{H_2SO_4,S}$ becomes larger than $C_{SO_3,S}$. Thus, for the conditions during the experiments (RH>0.3 %) this thermodynamic setup can be categorized as Case 2a.

With the Aspen Plus Databank pure–liquid saturation vapour pressure parameterization and $K_{H_2SO_4}=4.00\cdot10^9$ mol·kg$^{-1}$ and $^xK_{SO_3}=4.55\cdot10^{10}$ $C_{H_2SO_4,S}$ is always higher than $C_{SO_3,S}$ ($C_{H_2SO_4,S}=3.33$ µg·m$^{-3}$ and $C_{SO_3,S}=2.28$ µg·m$^{-3}$ at $a_w=2\cdot10^{-4}$ and N:S=0). Thus, this model setup can be also classified as Case 2a.

[Figure]

**Figure S7.I. (a)** The saturation concentration of $H_2SO_4$ $\left(C_{H_2SO_4,S}\right)$ and **(b)** $SO_3$ $\left(C_{SO_3,S}\right)$ in µg·m$^{-3}$ as a function of $a_w$ and $N{:}S$ at $T{=}288.8$ K. The $H_2SO_4$ dissociation equilibrium coefficients are $K_{H_2SO_4}{=}2.4{\cdot}10^9$ mol·kg$^{-1}$, and $^xK_{SO_3}{=}1.43{\cdot}10^{10}$. For the $H_2SO_4$ and $SO_3$ pure liquid saturation vapour pressures are used the N–K–L, Eq. (11) and Nickless, Eq. (12) parameterisations, respectively.

[Figure]

**Figure S7.II.** (a) The saturation concentration of $H_2SO_4$ $\left(C_{H_2SO_4,S}\right)$ and (b) $SO_3$ $\left(C_{SO_3,S}\right)$ in $\mu g \cdot m^{-3}$ as a function of $a_w$ and $N{:}S$ at $T{=}288.8$ K. The $H_2SO_4$ dissociation equilibrium coefficients are $K_{H_2SO_4}{=}4.00{\cdot}10^9$ $mol\ kg^{-1}$, and $^xK_{SO_3}{=}4.55{\cdot}10^{10}$. For the $H_2SO_4$ and $SO_3$ pure liquid saturation vapour pressures are used parameterisations from Que *et al.* (2011) (originally from the Aspen Plus Databank).

**S6 Geometrical mean diameter (GMD)**

[Figure]

**Figure S8. Measured and modelled GMD evolution as a function of (a) time and (b) RH for experiments 1 and 2 performed at**
**$T$=288.8 K. The modelled particles are composed of S(VI), $H_2O$ and $NH_3$ as a particle phase contaminant. The simulations correspond**
**to Case 1 with $H_2SO_4$ being the only evaporating S(VI) species, $K_{H_2SO_4}$=3.80·10⁹ mol·kg⁻¹, Case 2a with $H_2SO_4$ being the dominating**
**evaporating S(VI) species, $K_{H_2SO_4}$=4.00·10⁹ mol·kg⁻¹ and $^xK_{SO_3}$=4.55·10¹⁰, Case 2b with $SO_3$ being the dominating evaporating S(VI)**
**species, $K_{H_2SO_4}$ =5.00·10⁹ mol·kg⁻¹ and $^x K_{SO_3}$ =5.00·10⁹ and Case 3 with $SO_3$ being the only evaporating S(VI) species,**
**$K_{H_2SO_4}$=1.00·10¹¹ mol·kg⁻¹ and $^xK_{SO_3}$=5.00·10⁷ (see Supplement, Table S1, simulations 8–11 and 20–23). The pure liquid saturation**
**vapour pressures of $H_2SO_4$ and $SO_3$ are calculated with parameterizations from Que *et al.* (2011) (originally from the Aspen Plus**
**Databank).**

[Figure]

**Figure S9.** Measured and modelled GMD evolution as a function of (a) time and (b) RH for experiments 1 and 2 performed at $T=288.8$ K. The modelled particles are composed of S(VI), $H_2O$ and either NH3 or non–volatile, non–water–soluble organics as a particle phase contaminant for Case 2a, $K_{H_2SO_4}=2.40 \cdot 10^9$ mol·kg$^{-1}$ and $^x K_{SO_3}=1.43 \cdot 10^{10}$ (see Supplement, Table S1, simulation 2, 6, 14 and 17 ). The pure liquid saturation vapour pressures of $H_2SO_4$ and $SO_3$ are taken from N–K–L, Eq. (11) and Nickless, Eq. (12) parameterizations, respectively.

[Figure]

**Figure S10. Modelled and measured GMD evolution as a function of (a) time and (b) RH for experiments 1 and 2 performed at** *T=288.8* **K. The model results presented arise from Case 1 ($K_{H_2SO_4}$=2.00·10⁹ mol·kg⁻¹) without any particle phase contaminant. The pure liquid saturation vapour pressure of H₂SO₄ is calculated with Eq. (11), N–K–L parameterisation.**

Figure S11 compares the modelled and measured GMD evolution for experiments 1 and 2 performed at $T=288.8$ K when we use the data from Giauque et al. (1960) and Eq. (15) to derive the $H_2SO_4$ activity. $H_2SO_4$ is assumed to be the only evaporating S(VI) species (Case 1), the particle phase contamination consists of non–volatile non–water–soluble organics, and the pure–liquid saturation vapour pressure of $H_2SO_4$ is calculated with Eq. (11) or with the Aspen Plus Databank parameterization (see

5   Supplement, Table S1, simulations 5, 12, 19 and 24 in Table 2). The modelled GMD shrinkage agrees very well with the observations from experiments 1 and 2 when we use the tabulated $H_2SO_4$ chemical potential from Giauque *et al*. (1960) in combination with the pure–liquid saturation vapour pressure from the N–K–L parameterisation, Eq. (11). However, when we use the pure–liquid saturation vapour pressure parameterisation from the Aspen Plus Databank, the modelled particles evaporate earlier (at higher RH) than the observed particles. This due to ASPEN parameterisation which gives higher saturation

10  vapour pressures compared to N–K–L parameterization.

[Figure]

**Figure S11. Measured and modelled GMD evolution as a function of (a) time and (b) RH for experiments 1 and 2 performed at**
15  ***T=288.8* K. The modelled particles are composed of S(VI), $H_2O$ and non–volatile, non–water–soluble organics as a particle phase contaminant for Case 1 (see Supplement, Table S1, simulations 5, 12, 19 and 24). The pure liquid saturation vapour pressure of $H_2SO_4$ is calculated with Eq. (11), N–K–L parameterisation or with parameterisation from the Aspen Plus Databank. The $H_2SO_4$ activity is calculated with Eq. (15) using the tabulated chemical potentials from Giauque et al. (1960).**

[Figure]

**Figure S12. Measured and modelled GMD evolution as a function of (a) time and (b) RH for experiment 3 performed at a temperature range from 268 K to 293 K. The modelled particles are composed of S(VI), H₂O and NH₃ as a particle phase contaminant. The simulations correspond to Case 1, $K_{H_2SO_4}$ =3.80·10⁹ mol·kg⁻¹, Case 2a, $K_{H_2SO_4}$ =4.00·10⁹ mol·kg⁻¹ and ${}^xK_{SO_3}$=4.55·10¹⁰, Case 2b, $K_{H_2SO_4}$=5.00·10⁹ mol·kg⁻¹ and ${}^xK_{SO_3}$=5.00·10⁹ and Case 3, $K_{H_2SO_4}$=1.00·10¹¹ mol·kg⁻¹ and ${}^xK_{SO_3}$=5.00·10⁷ (see Supplement, Table S1, simulations 50-53). The pure liquid vapour pressures of H₂SO₄ and SO₃ are taken from Que et al., (2011) (original source Aspen Plus Databank).**

**Supplementary references**

Dawson, B. S. W., Irish, D. E., and Toogood, G. E.: Vibrational spectral studies of solutions at elevated temperatures and pressures. 8. A Raman spectral study of ammonium hydrogen sulfate solutions and the hydrogen sulfate–sulfate equilibrium, *The Journal of Physical Chemistry*, *90* (2), 334–341, 1986**.**

5    Giauque, W. F., Hornung, E. W., Kunzler, J. E., and Rubin, T. R.: The thermodynamic properties of aqueous sulphuric acid solutions from 15 to 300 K, J. Am. Chem. Soc., 82, 62–70, 1960.

Kulmala, M. and Laaksonen, A.: Binary nucleation of water–sulfuric acid system: Comparison of classical theories with different $H_2SO_4$ saturation vapor pressures, J. Chem. Phys., 93, 696–701, 1990.

Massoli, P., Lambe, A. T., Ahern, A. T., Williams, L. R., Ehn, M., Mikkila, J., Canagaratna, M. R., Brune, W. H., Onasch, T.

10   B., Jayne, J. T., Petaja, T., Kulmala, M., Laaksonen, A., Kolb, C. E., Davidovits, P., and Worsnop, D. R.: Relationship between aerosol oxidation level and hygroscopic properties of laboratory generated secondary organic aerosol (SOA) particles, Geophys. Res. Lett., 37, 5, L24801, 2010.

Myhre, C. E. L., Christensen, D. H., Nicolaisen, F. M., Nielsen, C. J. Spectroscopic Study of Aqueous $H_2SO_4$ at Different Temperatures and Compositions: Variations in Dissociation and Optical Properties. J. Phys. Chem. A, 107, 1979–1991, 2003.

15   Nickless, G.: Ed. "Inorganic Sulfur Chemistry", Elsevier, Amsterdam, 1968.

Noppel, M., H. Vehkamäki, and M. Kulmala, An improved model for hydrate formation in sulfuric–acid water nucleation, J. Chem. Phys, 116, 218–228, 2002.

Petters, M. D. and Kreidenweis, S. M.: A single parameter representation of hygroscopic growth and cloud condensation nucleus activity, Atmos. Chem. Phys., 7, 1961–1971, 2007.

20   Que, H., Song, Y., and Chen, C.: Thermodynamic modeling of the sulfuric acid–water–sulfur trioxide system with the symmetric Electrolyte NRTL model. J. Chem. Eng. Data, 56, 963–977, 2011.

Staples, B. R.: Activity and Osmotic Coefficients of Aqueous Sulfuric Acid at 298.15 K, J. Phys. Chem. Ref. Data, 10, 779–798, 1981.

Sullivan, R. C., Petters, M. D., DeMott, P. J., Kreidenweis, S. M., Wex, H., Niedermeier, D., Hartmann, S., Clauss, T.,

25   Stratmann, F., Reitz, P., Schneider, J., and Sierau, B.: Irreversible loss of ice nucleation active sites in mineral dust particles caused by sulphuric acid condensation, Atmos. Chem. Phys., 10, 11471–11487, 2010.

Topping, D. O., McFiggans, G. B., and Coe, H.: A curved multicomponent aerosol hygroscopicity model framework: Part 2–Including organic compounds, Atmos. Chem. Phys., 5, 1223–1242, 2005.

Zuend, A., Marcolli C., Booth , A. M., Lienhard, D. M., Soonsin, V., Krieger, U. K., Topping, D. O., McFiggans G., Peter, T.,

30   and Seinfeld, J. H.: New and extended parameterization of the thermodynamic model AIOMFAC: calculation of activity coefficients for organic–inorganic mixtures containing carboxyl, hydroxyl, carbonyl, ether, ester, alkenyl, alkyl, and aromatic functional groups, Atmos. Chem. Phys., 11, 9155–9206, 2011.